

# Ecophysiological modeling of the climate imprint on photosynthesis and carbon allocation to the tree stem in the North American boreal forest

Fabio Gennaretti[1], Guillermo Gea-Izquierdo[2], Etienne Boucher[3], Frank Berninger[4], Dominique Arseneault[5], Joel Guiot[1]

[1]CEREGE, Aix-Marseille University, CNRS, IRD, Aix en Provence, 13545, France
[2]Departamento de Sistemas y Recursos Forestales, CIFOR-INIA, Madrid, 28040, Spain
[3]Département de géographie, Université du Québec à Montréal, Montréal, H3C3P8, Canada
[4]Department of Forest Sciences, University of Helsinki, Helsinki, 00014, Finland
[5]Département de biologie, chimie et géographie, Université du Québec à Rimouski, Rimouski, G5L3A1, Canada

*Correspondence to*: Fabio Gennaretti (gennaretti@cerege.fr)

**Abstract.** A better understanding of the coupling between photosynthesis and carbon allocation in the boreal forest, with implicated environmental factors and mechanistic rules, is crucial to accurately predict boreal forest carbon stocks and fluxes, which are significant components of the global carbon budget. Here we adapted the MAIDEN ecophysiological forest model to better consider important processes for boreal tree species, such as non-linear acclimation of photosynthesis to temperature changes, canopy development as a function of previous year climate variables influencing bud formation, and temperature dependence of carbon partition in summer. We tested these modifications in the eastern Canadian taiga using black spruce (*Picea mariana* (Mill.) B.S.P.) gross primary production and ring-width data. MAIDEN explains 90% of the observed daily gross primary production variability, 73% of the full spectrum of the annual ring width variability and 20-30% of its high frequency component. The positive effect on stem growth due to climate warming in the last decades is well captured by the model. In addition, we illustrate the improvement achieved with each introduced model adaptation and compare the model results with those of linear response functions. This shows that MAIDEN simulates robust relationships with the most important climate variables (those detected by classical response-function analysis), and is a powerful tool for understanding how environmental factors interact with black spruce ecophysiology to influence present-day and future boreal forest carbon fluxes.

## 1 Introduction

Photosynthetic production is the primary motor of growth of trees and other vegetation. However, empirical studies have shown that correlation between photosynthetic production and diameter growth of trees is far from being perfect (Gea-Izquierdo et al., 2014; Rocha et al., 2006; Berninger et al., 2004). This is due to the fact that plant hydraulics (e.g. turgor pressure) and thermal limitations during very short periods of time can be more important than carbon (C) availability for





tree secondary growth (Kirdyanov et al., 2003; Rossi et al., 2016; Zweifel et al., 2016; Fatichi et al., 2014). These factors influence the proportion of net primary productivity allocated to stem growth each year, dampening the correlation between gross primary production (GPP) and growth. A better understanding of these factors and of carbon allocation mechanisms is needed when studying forest dynamics, forest carbon balance and the impact of climate change on forests. Indeed, carbon

allocated in different tree compartments (e.g. canopy, stem) has a specific function and is stored for a different length of time.

The varying roles of allocation and photosynthetic production are integrated in ecophysiological models (Li et al., 2014). Such models are important tools to analyze the direct influence of climate and other environmental factors (e.g. $CO_2$ concentration) on tree growth and biogeochemical processes in forest ecosystems (Li et al., 2016). Climate-growth

relationships have traditionally been assessed using empirical response functions based on linear relationships, thus considering the underlying processes as a black box. In contrast, ecophysiological models are built on mechanistic rules and allow considering non-stationarity and non-linearity in tree responses to environmental variables as well as their interactions (Vaganov et al., 2006). Ecophysiological models may be refined using model-data fusion approaches and optimization techniques (Guiot et al., 2014).

Different models with a different degree of ecophysiological complexity and/or spatio-temporal resolution have already been used to investigate the influence of climate and weather on tree growth in the boreal forest. Some studies focused on the drivers of photosynthetic capacity. For example, Mäkelä et al. (2004) proposed a model to study the influence of temperature on the seasonal variation in photosynthetic production of Scots pine through a delayed dynamic response. Other studies focused on the drivers of carbon allocation. For example, in Manitoba, Canada, a model related GPP and carbon allocation to

absorbed photosynthetically active radiation as a function of environmental constraints (Girardin et al., 2008). Another model, called CASSIA (Schiestl-Aalto et al., 2015), was developed to investigate how environmental factors and the ontogenetic stage of tree development influence the annual course of carbon sink–source dynamics in Scots pine stands. However, in spite of recent progress few models have been able to simultaneously simulate the meteorological control on daily photosynthetic production and the meteorological and phenological controls on daily carbon allocation for temperature-

limited boreal forest ecosystems. Modeling carbon assimilation and allocation in these ecosystems should consider specific mechanisms: (i) delayed response of photosynthesis to temperature (Gea-Izquierdo et al., 2010; Mäkelä et al., 2004); (ii) influence of preceding season conditions on current year canopy development (Salminen and Jalkanen, 2005); (iii) strong positive relationship between wood biomass production and temperature (Cuny et al., 2015).

Here we try to fill this gap by adapting the MAIDEN forest ecophysiological model, developed for temperate and

Mediterranean environments (Misson, 2004; Gea-Izquierdo et al., 2015), to mimic how weather and climate influence photosynthesis, phenology and carbon allocation in the North American boreal forest on a daily basis. MAIDEN offers an ideal framework to analyze the impact of introducing in the model relevant processes for carbon assimilation and allocation in temperature sensitive boreal trees. Firstly, we test and optimize these new model features on GPP and growth data from black spruce (*Picea mariana* (Mill.) B.S.P.), the dominant tree species across the North American boreal biome. Secondly,





we show the impact of single processes in the model runs and the improvements achieved with the new model adaptations. Lastly, we compare the simulated GPP and stem growth results with those obtained with conventional empirical linear response functions. With this comparison, we verify if the process-based ecophysiological model satisfactorily reproduces the variability of the observed data and if its simulations keep robust relationships with the most significant climate

variables.

## 2 Materials and Methods

### 2.1 The MAIDEN model

MAIDEN (Misson, 2004; Gea-Izquierdo et al., 2015) is a process-based forest ecophysiological model, which simultaneously simulates the course of photosynthesis and sets different phenological phases to determine the allocation of

carbon to different plant compartments in a dynamical manner. MAIDEN is able to consider the influence of several environmental factors on the forest water and carbon cycles. Starting from daily minimum-maximum air temperature, precipitation and $CO_2$ atmospheric concentration, MAIDEN models the phenological and meteorological controls on GPP and carbon allocation (Fig. 1). It explicitly allocates carbon to different pools (storage, canopy, roots and stem) on a daily basis using phenology-dependent mechanistic rules. It can also simulate the fractionation of carbon and oxygen isotopes to

estimate the isotopic ratios in tree-ring cellulose when the MAIDENiso isotopic modules are activated (Danis et al., 2012). Used in a forward mode, MAIDEN allows studying the effects of current and future global change on forest growth. The model has already been successfully optimized for *Quercus petraea* (Matt.) Liebl. and 12 Mediterranean species, including several *Pinus* spp. *and Quercus* spp. (Gaucherel et al., 2008a; Danis et al., 2012; Misson, 2004; Misson et al., 2004; Boucher et al., 2014; Gea-Izquierdo et al., 2015; Gaucherel et al., 2008b; Gea-Izquierdo et al., 2016). Drought and water stresses are

well taken into account, while the model has never been used for simulating tree growth in environments mostly sensitive to cold temperatures.

MAIDEN requires the definition of species and site dependent parameters (Misson, 2004; Gea-Izquierdo et al., 2015), such as soil texture and depth and the root to leaf mass fraction in the studied trees. The parameters that could not be set for the studied black spruce sites were analyzed with a sensitivity analysis and the most influential of them were estimated with

Bayesian optimization algorithms (Robert, 1996) using observed time series (daily GPP and annual ring width) as a reference. In total, six parameters influencing the GPP for black spruce and 12 parameters controlling the carbon allocation to the stem (Dstem) were optimized (they are described in the following paragraphs and in Table 1). The optimization was based on Markov Chain Monte Carlo (MCMC) sampling which, through its iterations, only retains combinations of parameters satisfying some conditions (Supplement S1; Fig. S1). Among the retained blocks of parameters, one block of six

parameters controlling GPP ("Plausible Block GPP") and one block of 12 parameters controlling Dstem ("Plausible Block Stem") were selected to illustrate the results with likely parameter values (Supplement S1).



### 2.1.1 Modeling GPP of boreal forests

In MAIDEN, daily stand GPP (g C m$^{-2}$ day$^{-1}$) is derived from the modeling of the coupled photosynthesis-stomatal conductance system. Leaf photosynthesis is calculated following De Pury and Farquhar (De Pury and Farquhar, 1997), while stomatal conductance is estimated using a modified version of the Leuning equation (Leuning, 1995; Gea-Izquierdo et al.,

2015). The photosynthesis-stomatal conductance system is estimated separately for sun and shade leaves based on the photosynthetic photon flux density they receive. The partition of leaf area index (LAI) in its shaded and sunlit fractions and the transmission and absorption of photosynthetically active radiation (PAR) are computed as explained by Misson (Misson, 2004), following De Pury and Farquhar (De Pury and Farquhar, 1997). After a sensitivity analysis, and as stated in the literature for boreal forests (Gea-Izquierdo et al., 2010; Mäkelä et al., 2004; Mäkelä et al., 1996), we found that the modeling

of assimilation/photosynthesis for black spruce is very sensitive to the parameters controlling the temperature dependence of maximum carboxylation rate ($Vcmax$; umol C m$^{-2}$ of leaves s$^{-1}$) and the water stress level ($\theta g$) influencing the stomatal conductance and consequently the intercellular $CO_2$ concentration. Following Gea-Izquierdo et al. (2015), $Vcmax$ is modeled as:

$$Vcmax_i = \frac{Vmax}{1+\exp(Vb \cdot (Tday_i - Vip))} \tag{1}$$

$Vcmax$ is a logistic function determining how daytime temperature ($Tday$; °C) controls the maximum carboxylation rate at the day $i$ if Rubisco is saturated. The parameters $Vmax$, $Vb$ and $Vip$ are the asymptote, the slope and the inflection point of $Vcmax_i$, respectively. In the model, the temperature dependence when the photosynthesis is instead limited by electron transport ($Jmax$) is considered as linearly related to $Vcmax$.

Following Gea-Izquierdo et al. (2015), $\theta g$ influencing stomatal conductance is modeled as:

$$\theta g_i = \frac{1}{1+\exp(soilb \cdot (SWC_i - soilip))} \tag{2}$$

$\theta g$ is a logistic function, which varies from 0 (maximum stress) to 1 (no stress) at the day $i$ depending on the soil water content ($SWC$; mm). $soilb$ and $soilip$ are the slope and the inflection point of $\theta g_i$, respectively.

With its already published MAIDEN configuration (Gea-Izquierdo et al., 2015), the model overestimated black spruce GPP in spring. This is due to the fact that the model has been developed for temperate-Mediterranean trees where it can be

assumed no time delay between photosynthesis and temperature increase (i.e. no temperature acclimation). However, such a delay is common in boreal trees (Gea-Izquierdo et al., 2010; Mäkelä et al., 2004). For this reason, we modified MAIDEN by including an extra function and an extra parameter ($\tau$) to take into account acclimation of photosynthesis to temperature. Basically, we replaced $Tday$ in Eq. (1) by a temperature transformation ($S$), which responds smoothly with a determined time lag to temperature variations. $S$ of the day $i$ was computed from the following differential equation (Mäkelä et al.,

2004), which was solved with the Euler's method:

$$\frac{dS_i}{di} = \frac{Tday_i - S_i}{\tau} \tag{3}$$



The new parameter $\tau$ is a time constant interpretable as the number of days needed by the photosynthetic apparatus to acclimate to changing temperature.

### 2.1.2 Modeling carbon allocation to the stem (Dstem) in boreal forest

MAIDEN allocates the daily available carbon from photosynthesis and stored non-structural carbohydrates to all plant
compartments (stem, roots, canopy and storage) using functional rules specific to each of the five phenological phases characterizing a year (see Fig. 1). Although we maintained the original MAIDEN structure, we modified some previously used functional rules from Gea-Izquierdo et al. (2015) to consider significant processes for the boreal forest. We describe below the functional rules controlling Dstem, according to phenological phases.

During the "winter period 1" (phase 1) few processes are active. However, at the beginning of each year, the model defines
the maximum amount of carbon that the canopy can potentially contain that year ($AlloCcanopy_j$; g C m$^{-2}$ of stand) as a function of previous year climate variables. Based on a correlation analysis (see our results below) and on previous studies on black spruce forests (Girardin et al., 2016; Ols et al., 2016; Mamet and Kershaw, 2011), we modified the model to consider the effect of the previous year July-August temperature and April precipitation in replacement of the mean soil water content for May–December of the previous year such as in Gea-Izquierdo et al. (2015). Previous year climate
conditions of specific months are known to influence shoot extension of boreal trees likely because they control accumulation of resources in the buds (Salminen and Jalkanen, 2005). Here, we calculated the carbon potentially allocated each year to the canopy with the following equations:

$$CanopyMult = \frac{1}{1+exp\,(CanopyT \cdot Temp_{j-1})} \cdot \frac{1}{1+exp\,(CanopyP \cdot Precip_{j-1})} \qquad (4)$$

$$AlloCcanopy_j = 0.7 \cdot MaxCcanopy + 0.3 \cdot MaxCcanopy \cdot CanopyMult$$

Where $Temp_{j-1}$ is the previous year mean July-August temperature (detrended and transformed to z-scores), $Precip_{j-1}$ is the previous year April precipitation (detrended and transformed to z-scores), and $MaxCcanopy$ is the absolute maximum canopy carbon reservoir according to forest traits, diameter distributions and previously published allometric equations (Chen, 1996; Bond-Lamberty et al., 2002a; Bond-Lamberty et al., 2002b). $CanopyT$ and $CanopyP$ are two parameters that were optimized and representing the slopes of the relationships between $CanopyMult$ (i.e. the overall climate dependence)
and $Temp_{j-1}$ or $Precip_{j-1}$, respectively.

During the "winter period 2" (phase 2), growing degree days (GDD) start to accumulate. We computed accumulation of GDD by summing the mean daily temperature values over 3°C (Nitschke and Innes, 2008; Man and Lu, 2010). MAIDEN simulates budburst (i.e. the transition from the phenological phase 2 to 3) either when the GDD sum threshold is reached (parameter $GDD1$) or when a selected day of the year related to photoperiod is passed (parameter $vegphase23$). With this
model configuration, the start of the growing season overreacted to GDD yearly variations and thus, the correlation between yearly stem carbon allocation and May temperature was much higher than that between the ring width observed data and May temperature. To correct this simulated bias, we modified MAIDEN by adding a mechanism reducing the inter-annual





variability of budburst dates. This mechanism simulates the acclimation of the plants to varying GDD sums from year to year. Basically, the yearly time series of days of the year corresponding to budburst (determined by GDD and photoperiod) is smoothed at the beginning of each simulation with a *n*-year cubic smoothing spline. The integer number *n* was called *day23_flex* and optimized like the other parameters.

The "budburst phase" (phase 3) starts with budburst and ends when $AlloCcanopy_j$ is reached or when the carbon in the storage reservoir (i.e. stored non-structural carbohydrates) is lower than a minimum value (Misson, 2004). Here, this phase was set to be shorter than 51 days, based on available spruce budburst and shoot elongation data (Lemieux, 2010). During this phase, the daily available carbon ($CT_i$) comes from photosynthesis and mobilization of storage carbon. The parameter *Cbud*, which was optimized, is the amount of storage carbon that is used each day by the plant. The total $CT_i$ amount is then

allocated to the canopy, the roots or the stem following some functional rules. In the previous version of MAIDEN (Gea-Izquierdo et al., 2015), these rules were functions of daily soil moisture and temperature. Here these rules did not improve the simulated results and we retained a simpler version independent from climate:

$$Cstem_i = CT_i \cdot (1 - h3) \tag{5}$$

where $Cstem_i$ is the portion of $CT_i$ allocated to stem and $h3$ is a parameter to be defined in the range between 0 and 1. The

rest of $CT_i$ is allocated to the canopy or the roots, respecting a prescribed 1.65 root to canopy mass ratio (Czapowskyj et al., 1985; Jenkins et al., 2003).

During the "growth and accumulation phase in summer" (phase 4), $CT_i$ comes only from the photosynthesis and is allocated either to stem growth or storage as a function of climate forcing. In the previous version of MAIDEN for water limited sites (Gea-Izquierdo et al., 2015), the allocation rule used a combination of daily soil moisture and temperature as predictors. Here

for temperature limited sites, we only used temperature and set the soil moisture part to be with a null effect (i.e. always equal to 1, note that for more water limited boreal sites this water stress dependence can be used):

$$Cstem_i = CT_i \cdot \left( 1 - 0.8 \cdot exp\left( -0.5 \left( \frac{Tmax_i}{st4temp} \right)^2 \right) \right) \tag{6}$$

where $Tmax_i$ is the daily maximum temperature, and $st4temp$ is a parameter that correspons to the inflection point of the function where roughly 50% of $CT_i$ is allocated to the stem.

The transition from phase 4 to the "fall phase" (phase 5) is determined by either the parameter *photoper* (threshold of duration of daylight in hours) or by the occurrence of negative minimum daily temperature values after the 1[st] of September. During the "fall phase", all photosynthetic products are allocated to the storage reservoir and mortality of fine roots occurs. No specific functional rule influences Dstem during this phase.

The equation controlling partial carbon losses from the canopy (i.e. litterfall) and thus influencing the photosynthetic

capacity in the studied evergreen species, runs all year round. This equation is inspired from Maseyk et al. (2008):

$$outCcanopy_i = (PercentFall \cdot AlloCcanopy_j) \cdot exp\left( -0.5 \left( \frac{DOY_i - 1}{OutMax} \right)^{OutLength} \right) -$$





$$\left(PercentFall \cdot AlloCcanopy_j\right) \cdot exp\left(-0.5\left(\frac{DOY_i}{OutMax}\right)^{OutLength6}\right) \qquad (7)$$

where $outCcanopy_i$ is the carbon loss from the canopy at day $i$ and is influenced by parameters $PercentFall$, $OutMax$, and $OutLength$ (to be optimized), which determine the yearly canopy turnover rate, the day of the year with maximum losses and the length of the period with losses, respectively.

**2.2 Study sites and data**

**2.2.1 Eddy covariance observations**

Eddy covariance stations provide measurements of net ecosystem production (NEP), as well as estimates of gross ecosystem production (GEP) and respiration (R) for specific sites. There is one eddy covariance station from the Fluxnet network located in a mature black spruce forest in the northern Quebec taiga. It is the "Quebec Eastern Old Black Spruce" station

(EOBS; 49.69N and 74.34W; Bergeron et al., 2007; http://fluxnet.ornl.gov/site/269) with data from 2003 to 2010. Although NEP and R from eddy covariance stations are not directly comparable with MAIDEN outputs because they integrate all ecosystem components (e.g. soil heterotrophic respiration), GEP was assumed to be comparable to the simulated GPP, because GEP only derives from the autotrophic components of the ecosystem (Gea-Izquierdo et al., 2014). Consequently, we integrated the GEP half-hourly time series from the EOBS site in a daily time series (Fig. 2), to make it comparable with the

MAIDEN GPP for the same location. We then used these data to optimize the six parameters influencing the stand GPP simulated by the model for black spruce forests. Hereafter, we will employ the term GPP also to denote GEP estimates from the EOBS site.

**2.2.2 Ring with data from the northern Quebec taiga**

We assumed that the yearly Dstem is proportional to tree-ring growth in order to use ring width data to optimize MAIDEN

(12 influential parameters). We used data from 46 black spruce trees sampled in the riparian forests of five lakes in the eastern Canadian taiga (Gennaretti et al., 2014; the coordinates of the central point are 54.26N and 71.34W). The time series (Fig. 3; Dataset S1) were standardized using a site-specific Regional Curve Standardization (Gennaretti et al., 2014) and averaged to obtain a regional chronology (hereafter RW) to compare with MAIDEN annual stem carbon increments (g C·m$^{-2}$ of stand·year$^{-1}$). However, because we wanted to analyze both the multi-decadal and the inter-annual variability of carbon

allocation, we subsequently detrended all standardized ring width time series by subtracting their respective 10-year cubic smoothing splines (50% frequency cutoff for 10-year periods). The resulting detrended regional chronology (hereafter RWhighF) was compared with MAIDEN annual stem carbon increments, detrended in the same way. RWhighF was also used as a reference for the optimization of the MAIDEN parameters (Supplement S1). We preferred to optimize MAIDEN on RWhighF values because tree-ring high frequencies are much more robust regionally across sites and trees than low

frequencies. Observed and simulated low frequencies were only compared after the optimization of the model parameters. MAIDEN outputs were simulated for the central point of the source area of ring width data over the 1950-2010 period. Such





a spatial aggregation of tree-ring data is known to reduce non-climatic noise at the site level, thus increasing the coherence between modeled and observed time series (Breitenmoser et al., 2014).

### 2.2.3 Climate data

MAIDEN needs daily climate data as inputs. These data were obtained from the gridded interpolated Canadian database of
daily minimum–maximum temperature and precipitation for 1950-2015 (Hutchinson et al., 2009; http://cfs.nrcan.gc.ca/projects/3/4). We extracted the time series from the grid cells nearest to the studied locations (the eddy covariance station and the central point of the region with collected ring width data). $CO_2$ atmospheric concentration values for the same sites were obtained from the nearest grid cell of the CarbonTracker measurement and modeling system (2000-2015 period; Peters et al., 2007; http://www.esrl.noaa.gov/gmd/ccgg/carbontracker/). In order to obtain longer $CO_2$ time
series (1950–2015), the seasonal cycle from the selected cells of the CarbonTracker grid was superimposed to the long-term $CO_2$ trend from the Mauna Loa observatory (1958-2015; Keeling et al., 1976; http://www.esrl.noaa.gov/gmd/ccgg/trends/) with removed seasonal cycle, extrapolated before 1958, and modified by adding the mean offset between the selected cells and the Mauna Loa cell in the CarbonTracker grid.

### 2.3 Response function analysis

Linear response functions are regression models used to quantify the proportion of the variability of the observed data (stem growth or GPP in our case) that can be explained by climate variables. These functions do not directly explore the mechanistic rules such as process-based models and are only optimized to achieve the best fit. Thus, comparing the results of linear functions and process-based models can help verify if the model performance is satisfactory and if some important climatic factor related to some process is missing in the model. We used linear response functions to analyze the
relationships between observed daily GPP at EOBS and daily mean, maximum and minimum temperatures or weekly precipitation (explored time lag from 0 to 30 days before; in the case of precipitation, lag *n* indicated the sum of the daily precipitation of the week ending in day *n*). In this analysis we excluded the winter days (days of the year between November 15[th] and April 1[st]) where GPP is zero. The 10 predictors most strongly correlated with GPP (and not highly correlated with each other; pairwise r∈[-0.8, 0.8]) were retained for the analysis. All linear response functions, resulting from a combination
of these 10 predictors, were tested and classed according to their Bayesian information criterion (BIC).

We also used linear response functions to analyze the relationships between RWhighF and climate variables (same methodology than for GPP). We tested as predictors all monthly temperature and precipitation values of the previous and current years. Time windows of 31 days were used to obtain the time series of monthly data (over the 1950-2010 period) for each day (central day), averaging the values of each window and each year. These climate time series were also detrended
such as RWhighF.



## 3. Results and Discussion

### 3.1 GPP and tree-ring growth variability explained by MAIDEN

The model explained a large proportion of the observed GPP daily variability (90%; r=0.95; Fig. 4a). The posterior distributions of the parameters were quite sharp (Fig. S2; Table 1). This means that the model posterior probability (i.e.

model plausibility) increased significantly with specific values of the selected parameters and these values were retained by the MCMC sampling. Although the model was optimized with daily data, the GPP time series also reproduced the annual variability of the observed data quite well (Fig. 5). However, the ensembles of daily and annual time series retained by the MCMC sampling were not always centered on the observed time series (Fig. 5). This reflects the fact that the MCMC sampling maximized the model plausibility according to the model structure and, by doing so retained similar blocks of

parameters. Thus, the range of simulated values in Fig. 5 obtained with all retained iterations should be interpreted as the uncertainty due only to parameter selection while the uncertainty due to the non-perfect fit between observations and simulations was not taken into account.

As expected, the ring growth variability at our sites was more linked to temperature than precipitation variables (see Fig. 6 and Gennaretti et al., 2014; Mamet and Kershaw, 2011; Nicault et al., 2014). The model explained about 20-30% of the

observed yearly RWhighF variability corresponding to correlations of 0.58-0.66 (Fig. 4b). This is a good result because simulated detrended annual GPP values (i.e. photosynthetic assimilation before any carbon allocation) had only negative $R^2$ with RWhighF (Fig. 4c; meaning performance worse than a straight line centered on RWhighF). This suggests that the modified MAIDEN daily partition of carbon in the plant compartments significantly improved the concordance with tree-ring observations. Again, the parameters' posterior distributions were quite sharp (Fig. S3; Table 1), meaning that the

MCMC sampling selected similar blocks of parameter resulting in high model posterior probabilities. The variance explained by the model increased importantly when the observed and simulated time series of stem growth were analyzed with their trends ($R^2$=0.73 and r=0.86; Fig. 7b). Indeed, the positive trend due to the warming of the last few decades was well captured by the model simulations of stem increments. This explained variance was higher to that explained by MAIDEN for Mediterranean sites ($R^2$ slightly above 0.5; Gea-Izquierdo et al., 2015).

### 3.2 Underneath mechanistic rules

The modeled impact of temperature on the maximum rate of Rubisco-catalyzed carboxylation ($Vcmax$) is shown in Fig. S4. This figure was obtained using Eq. (1) and (3) with the parameters of Plausible Block GPP and using actual temperature data. The obtained $Vcmax$ values were comparable to those obtained for another mature black spruce forest in Saskatchewan, Canada (Rayment et al., 2002). Furthermore, the impact of soil water content on the water stress level ($\theta g$)

influencing the stomatal conductance is shown in Fig. S5. Simulated GPP values were sensible to all single parameters controlling $Vcmax$ or $\theta g$, except *soilb* (Fig. S6-S10). The temperature transformation ($S$) introduced here in MAIDEN also influenced the simulation results (Fig. 8). With no time delay between photosynthesis and temperature increases (i.e. $\tau$ = 1





and $S = Tday$) MAIDEN overreacted to temperature variations in spring and the GPP annual cycle was antedated (start in spring and highest summer values were too early). In contrast, the use of S with $\tau$ values between 10 and 15 days synchronized the GPP annual cycle with observations. This means that black spruce photosynthetic capacity needs about 10-15 days to acclimate to higher daily temperature (e.g. $\tau$ equal to 12.43 was selected for Plausible Block GPP). This time

delay is a little longer than that previously found for black spruce but comparable to values found for Scots pine (Mäkelä et al., 2004; Gea-Izquierdo et al., 2010; Gea-Izquierdo et al., 2014).

We modified important processes for carbon allocation in order to adapt MAIDEN to black spruce. For example, previous year precipitation and temperature values influenced the potential maximum amount of carbon that the canopy can contain during the growing season as illustrated in Fig. 9 (see Eq. (4)). Basically, if both previous April precipitation and July-

August temperature indexes are negative, the potential amount of carbon simulated by the model would be maximum, otherwise it would be minimum. This was coherent with the correlations shown in Fig. 6 and we can propose the following reasons to explain this behavior: warm previous Aprils with infrequent late snowfalls may accelerate snowmelt and the start of the previous growing season, allowing optimal reserve accumulation during the previous year with repercussions on the following growing year. This mechanism may be significant especially if we do not observe high temperatures limiting soil

water availability and reserve accumulation during the previous summer (Girardin et al., 2016). It has already been shown that shoot elongation of boreal conifers is determined by climate conditions during bud formation (Salminen and Jalkanen, 2005). However, for Scots pine, previous summer temperatures are positively correlated with shoot elongation, while in our case, the opposite process was simulated. Clearly, we need more data on canopy development and shoot elongation to verify the model results.

Another important process is the start of the growing season. According to our simulations, the start could not happen later than June 17[th] (Figs. S3d and S11; Table 1) and was influenced by the GDD sum and the photoperiod, which are known to be relevant for the black spruce budburst along with the tree provenance (Rossi and Bousquet, 2014). However, because we added a mechanism to smooth yearly variations, more years were needed by the plants to acclimate to more or less fast GDD accumulations in winter-spring. With the parameters of Plausible Block Stem, the median onset of the growing season was

June 10[th] (similar to observations for black spruce in northern Manitoba, Canada; Bronson et al., 2009) with a standard deviation of 7.8 days. If the smoothed term was excluded, the standard deviation increases to 9.4 days (see Fig. S11a). The inclusion of the smoothed mechanism also decreased the correlation between the simulated detrended annual Dstem and May average temperature from 0.70 to 0.59. Although this is still a too high correlation, it was closer to the correlation between RWhighF and May temperature (r=0.27; Fig. 6). These results show how the new model configuration decreased

the yearly variability of the growth onset and helped achieving more plausible correlations with climate variables. According to the simulations, the onset of the growing season shifted by 7 days from June 14[th] to June 7[th] between the 1950-1970 and 1990-2010 periods (Fig. S11b-c). This result is consistent with the study of Bronson et al. (2009) on the effect of warming on black spruce budburst.





In phase 3, corresponding to Budburst, a portion of the available carbon simulated by MAIDEN comes from stored non-structural carbohydrates. In our case, this amount was quantified as about 1.69 g C·m$^{-2}$ of stand·day$^{-1}$ and the model appeared quite sensitive to this parameter as proved by the sharp posterior density (Fig. S3f). During phase 3 of our simulations, almost all available carbon was allocated to the canopy and roots ($h3 \approx 0.9905$; Eq. (5); Fig. S3g; Table 1). For this reason, the previously used soil moisture and temperature dependences, determining the portion of carbon allocated to the stem (Gea-Izquierdo et al., 2015), did not improve the results and could be excluded here. The partition of carbon during the growth and accumulation phase in summer (phase 4) was instead modeled as a function of temperature (Eq. (6)). Warmer temperature corresponded to a greater portion of carbon allocated to the stem (Fig. 10). These results are in part in line with the results of Cuny et al. (2015), showing that woody biomass production is low in the first part of the growing season for most coniferous tree species because it follows the seasonal course of temperature (highest peak in summer). The simulated accumulation of carbon to the stem ended each year when the photoperiod became shorter than about 13.41 hours (Fig. S3i; Table 1), corresponding to September 2$^{nd}$. The model was very sensitive to this parameter which is known to impact black spruce dormancy induction (D'aoust and Cameron, 1982).

Another important process for carbon allocation is the definition of the carbon losses from the canopy, a process that influences the seasonal course of the photosynthetic capacity. According to the simulations, the canopy mean turnover rate was about 13-14% (Fig. S3j; Table 1), and corresponded well to previously published values for boreal spruce species (Ťupek et al., 2015). The simulated annual cycle of losses (Fig. S12) culminated on July 2$^{nd}$ and 80% of the losses occurred between May 27$^{th}$ and July 19$^{th}$. This cycle is also similar to published results showing that the majority of litterfall ($\approx$ 80%) occurs in summer during needle growth for conifer species (Maseyk et al., 2008).

## 3.3 Comparison between MAIDEN and response functions

The comparison between MAIDEN simulations and linear response functions confirmed the quality of the simulated results with the process-based model and, in some cases, justified our modeling choices.

In the case of daily GPP, we were able to draw the following conclusions by the response function analysis (Table 2). First, MAIDEN performed better than response functions in explaining the variability of daily GPP ($R^2$=0.90 vs 0.69), suggesting that it properly simulates climate-driven processes governing photosynthetic assimilation. Second, most of the variance explained by the response functions was due to temperature variables, reflecting the greater sensitivity of northern black spruce forests to temperature as compared to drought stress (Gennaretti et al., 2014) and justifying the modeling in MAIDEN of the maximum carboxylation rate as a function of temperature. Third, only temperature variables of preceding days were retained, justifying the inclusion of our acclimation function of photosynthesis to temperature in order to increase the influence of previous days. Fourth, the coefficient estimate for precipitation of lag 0 (i.e. week ending in day 0) was negative, while the one of lag -2 was positive, even though these variables share 5 out of 7 days of data. The reduction of absorbed photosynthetically active radiation associated to cloudiness during raining days could explain this result.



In the case of annual Dstem, the selected response function (Table 3) captured 50% of the observed RWhighF variability using only three temperature variables. This, once again, confirms that black spruce forests in the study area are especially sensitive to temperature (Gennaretti et al., 2014). The MAIDEN simulated time series were able to respect the relationship with the significant monthly climate variables detected with the response function analysis. Indeed, correlation coefficients

of -0.39, 0.46 and 0.57 were obtained between MAIDEN Dstem and previous July-August, growing year July and growing year May-June temperature values, respectively (these coefficients are to compare with those in Table 3). This concordance supports the plausibility of the simulated series. However, the explained variability with the best response function (50%) was greater than with MAIDEN (20-30%; $r \approx 0.65$; Fig. 4b), suggesting that the process-based modeling can potentially be improved with additional data and including stronger legacy effects of the year preceding ring formation (Girardin et al.,

2016). Indeed, most of the variance explained by the response function was due to a negative correlation with the temperature of the previous summer. Contrasting correlations with summer temperature values of the previous and growing year are also visible in Fig. 6 and have already been observed for black spruce (Mamet and Kershaw, 2011; Ols et al., 2016).

## 4. Conclusion

In this study, we adapted a process-based forest ecophysiological model, developed for temperate and Mediterranean forests,

to accurately simulate GPP and Dstem for black spruce, the dominant species across the North American boreal forest. The used model, MAIDEN (Misson, 2004; Gea-Izquierdo et al., 2015), has the specificity to simultaneously simulate the course of photosynthesis and phenological phases characterized by specific allocation rules. The model was able to represent the tree-ring inter-annual variability even though RWhighF was poorly explained by the simulated annual GPP (Fig. 4b-c), suggesting that relationships between GPP and wood production are complex and non-linear (Rocha et al., 2006). Significant

simulation improvements were obtained introducing in the model important processes for temperature sensitive boreal forests, such as: (i) the acclimation of photosynthesis to temperature over several days (see Gea-Izquierdo et al., 2010; Mäkelä et al., 2004); (ii) the influence of previous year climate conditions affecting bud formation on the potential amount of carbon that the canopy can contain each year (see Salminen and Jalkanen, 2005); (iii) the positive relationship between temperature and Dstem in summer (see Cuny et al., 2015). Although we used black spruce data from the northern Quebec

taiga to test and optimize the model, the new model adaptations have the potential to work with other boreal regions and tree species. Indeed, the effect of the introduced functions can be amplified, reduced or canceled out in the Bayesian optimization procedure according to the relevance of the process in the studied forest (for example if $\tau$ in Eq. (3) is set to 1 the time delay between photosynthesis and temperature increases becomes zero again). Furthermore, using Bayesian optimization algorithms, we can analyze the posterior distribution (i.e. uncertainty) of each optimized parameter.

Although the simulated results are satisfactory, we have to consider two important limits and error sources of the study. First, for the optimization of carbon allocation, we assumed that stem biomass (or carbon) increments were proportional to ring growth. This was necessary because data from field plots were not available from all study sites. A recent study showed that



the maximum rate of ring width increase during the growing season precedes the maximum rate of increase in wood biomass and that these processes could exhibit differential sensitivities to local environmental conditions (Cuny et al., 2015). However, Cuny et al. (2015) also highlighted that wood biomass production follows the seasonal course of temperature in coniferous forests and this is exactly what we got once MAIDEN was optimized. Indeed, almost all available carbon in spring was allocated to the canopy and roots (Fig. S3g; Table 1) and Dstem in summer increased with temperature (Fig. 10). Furthermore, the used ring width series were highly correlated with July-August temperature as expected for wood biomass production. Second, we modeled GPP and carbon stem increments of a boreal tree species using mechanistic rules which increased the capability of MAIDEN to reproduce observed variations. However, our choice of mechanistic rules was subjective in part and depended on previous physiological knowledge and on model-data comparisons. Such model refining is an important step of all model-data fusion approaches (Guiot et al., 2014) and increases our understanding of ecosystem functioning and responses. Nevertheless, the proposed mechanistic rules should be verified in the future with additional data from a wider boreal area.

Boreal ecosystems are crucial carbon stores that must be urgently quantified and preserved (Bradshaw et al., 2009). Their future evolution is extremely important for the global carbon budget. Development of process-based models, such as the one used and improved here, combined with continuous field data acquisition, will help disentangle the role of the different environmental factors and underneath mechanisms on present and future boreal forest carbon fluxes. In this context, we believe that our study helps to understand how boreal forests assimilate and allocate carbon depending on weather/climate conditions.

## Acknowledgements

This project has received funding from the European Union's Horizon 2020 research and innovation programme under the Marie Sklodowska-Curie grant agreement No 656896. The CRD NSERC ARCHIVES project funded the sampling of the tree-ring data. We acknowledge all institutes and persons providing the other used data: Natural Resources Canada for the climate data; NOAA Earth System Research Laboratory for the $CO_2$ data; the Fluxnet project and Hank Margolis (Université Laval) for the eddy covariance data, Gabriel Rodrigue for the soil data at our tree-ring sites.

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





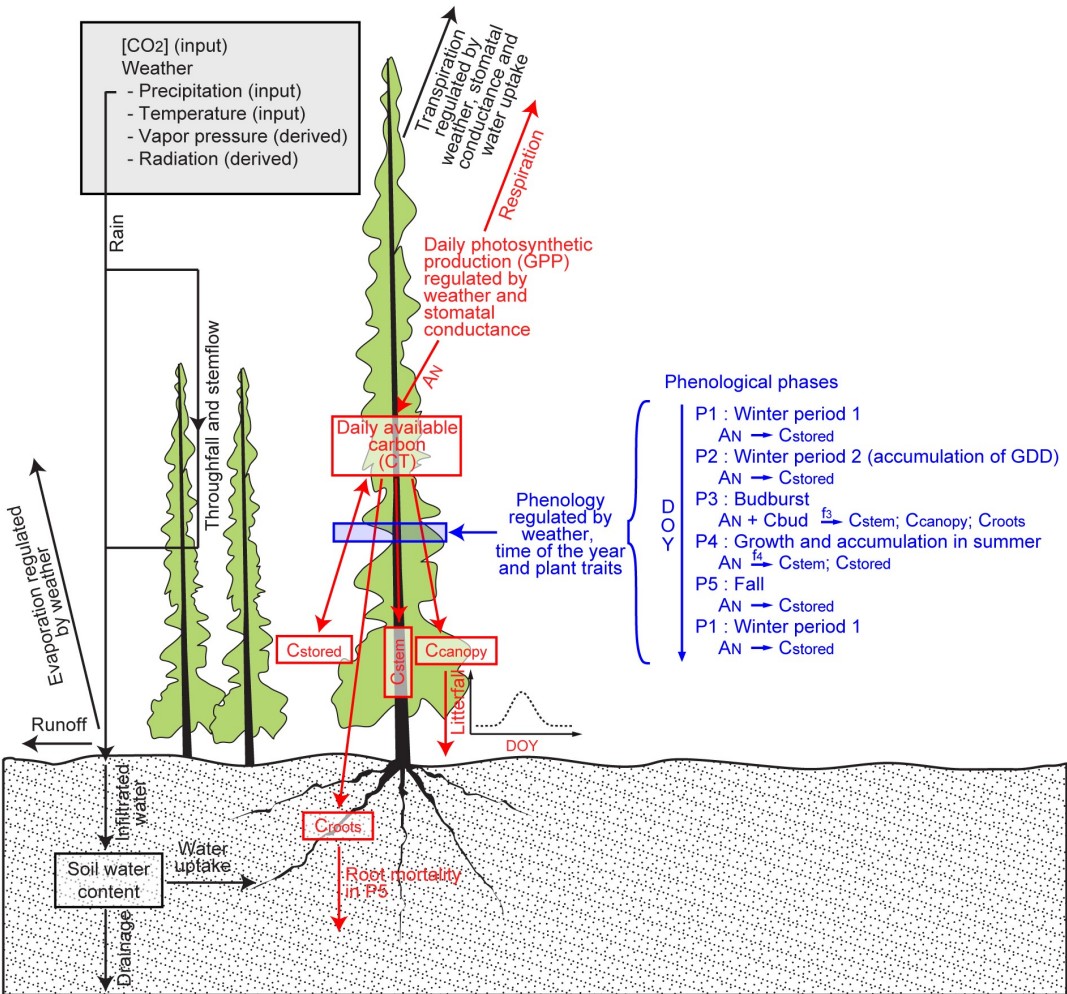

**Figure 1: MAIDEN simulated phenology (blue), water (black) and carbon (red) fluxes. AN: net photosynthesis corresponding to net primary production. Cstored, Cstem, Ccanopy, Croots: carbon allocated daily to stored non-structural carbohydrates, stem, canopy or roots. DOY: day of the year (1-365). GDD: growing degree days. f3 and f4: functions determining carbon allocation in**
5   **phase 3 and 4. Cbud: amount of storage carbon that is used each day by the plant in phase 3.**





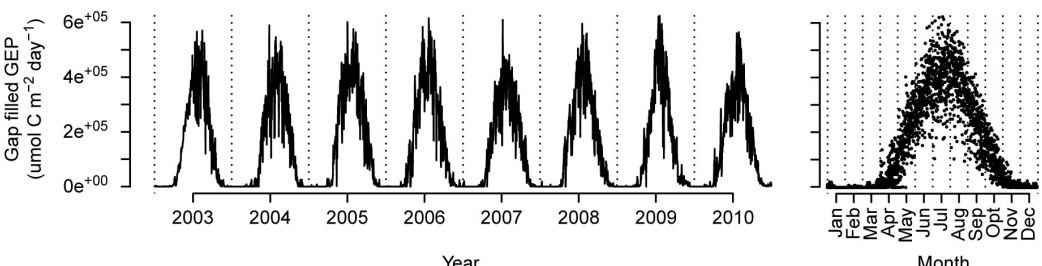

**Figure 2: "Quebec Eastern Old Black Spruce" (EOBS) GPP gap filled daily time series (left) and annual cycle (right). 2003-2010 period. Units are umol C m$^{-2}$ day$^{-1}$.**

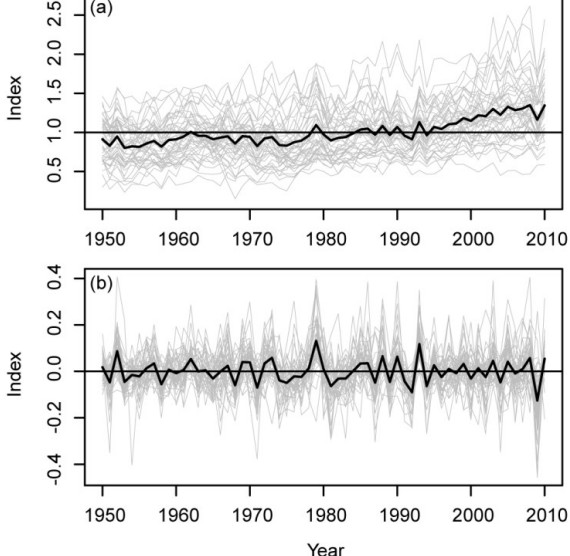

5    **Figure 3: Time series of used ring width data. (a) Site-specific RCS standardized. (b) Site-specific RCS standardized and detrended. Individual series are in grey and the mean (corresponding to RW and RWhighF) is in black.**





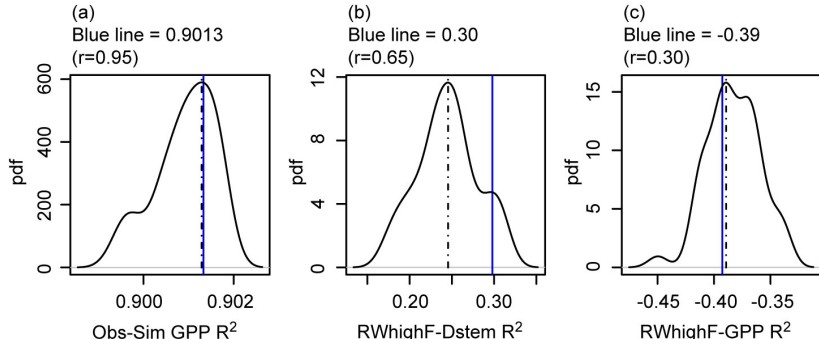

**Figure 4: Variance explained by the model. (a) $R^2$ between observed and simulated GPP daily values. $R^2$ (computed on data transformed to z-scores) between the mean of the detrended series of black spruce ring growth (RWhighF) and simulated yearly detrended C allocation to the stem (b) or GPP (c). Vertical dashed line is the mode and blue line is the value with Plausible Block GPP (in (a)) or with Plausible Block Stem (in (b) and (c)). All pdfs are based on 50 simulations.**

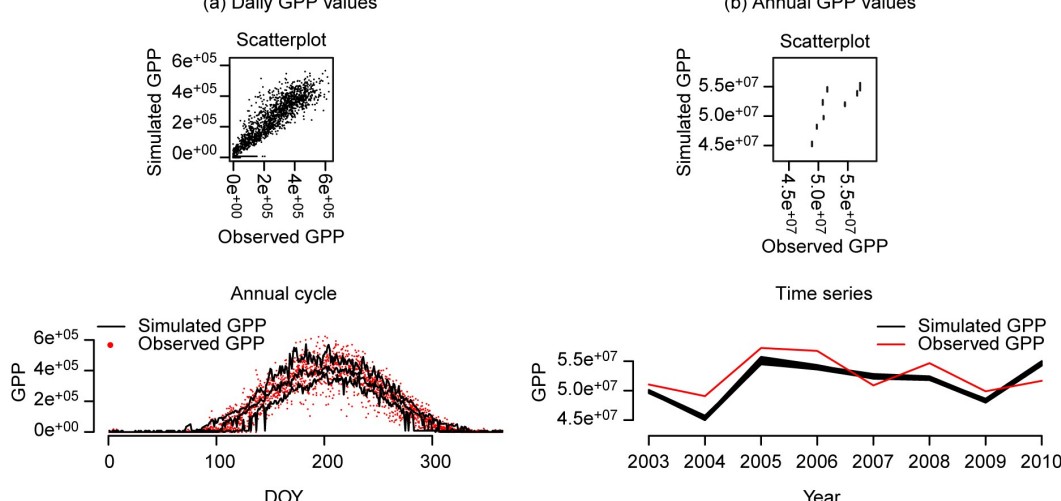

**Figure 5: Comparison between observed GPP values and MAIDEN simulated values at the Quebec Eastern Old Black Spruce site. (a) Daily values (units are umol C m$^{-2}$ day$^{-1}$). In the scatterplot ($R^2$=0.90; r=0.95), observations are compared with the values obtained with Plausible Block GPP. In the annual cycle plot, black lines are the medians, the 5$^{th}$ and the 95$^{th}$ percentiles of the simulated values from all iterations retained by the MCMC sampling. (b) Annual values (units are umol C m$^{-2}$ year$^{-1}$). In both plots, observations are compared with the values from all iterations retained by the MCMC sampling. In the scatterplot, the $R^2$ of the data is 0.31 (r=0.76).**





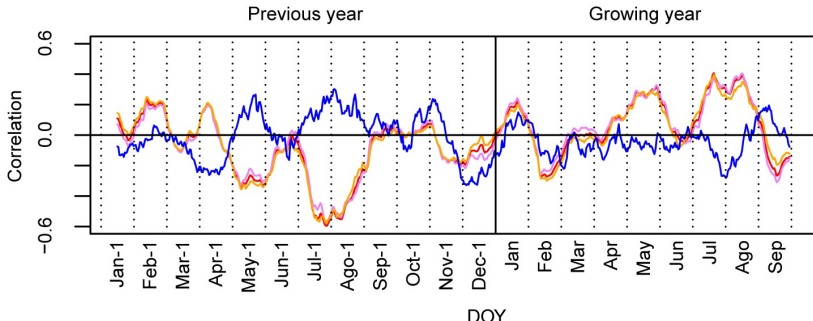

**Figure 6: Correlation between the mean of the detrended series of black spruce ring growth (RWhighF) and monthly climate variables of the study area (precipitation in blue and mean, maximum and minimum temperature in red, violet and orange respectively). For the climate variables, time windows of 31 days are used to obtain time series of monthly data (over the 1950-2010 period) for each day (central day), averaging the values of each window and each year. These climate time series are then also detrended.**

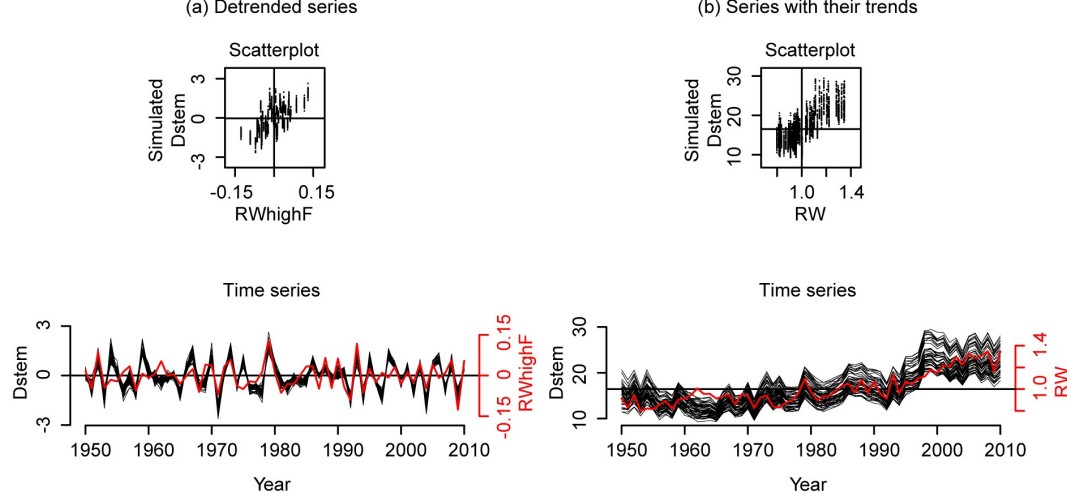

**Figure 7: Comparison between the observed mean series of black spruce ring growth (growth indexes) and MAIDEN simulated carbon allocation to stem (g C m$^{-2}$ year$^{-1}$). (a) Detrended series (in the scatterplot the R$^2$ computed on data transformed to z-scores is 0.24; r=0.62). (b) Series with their trends (in the scatterplot the R$^2$ is 0.73; r=0.86). In all plots, observations are compared with the values from all iterations retained by the MCMC sampling.**





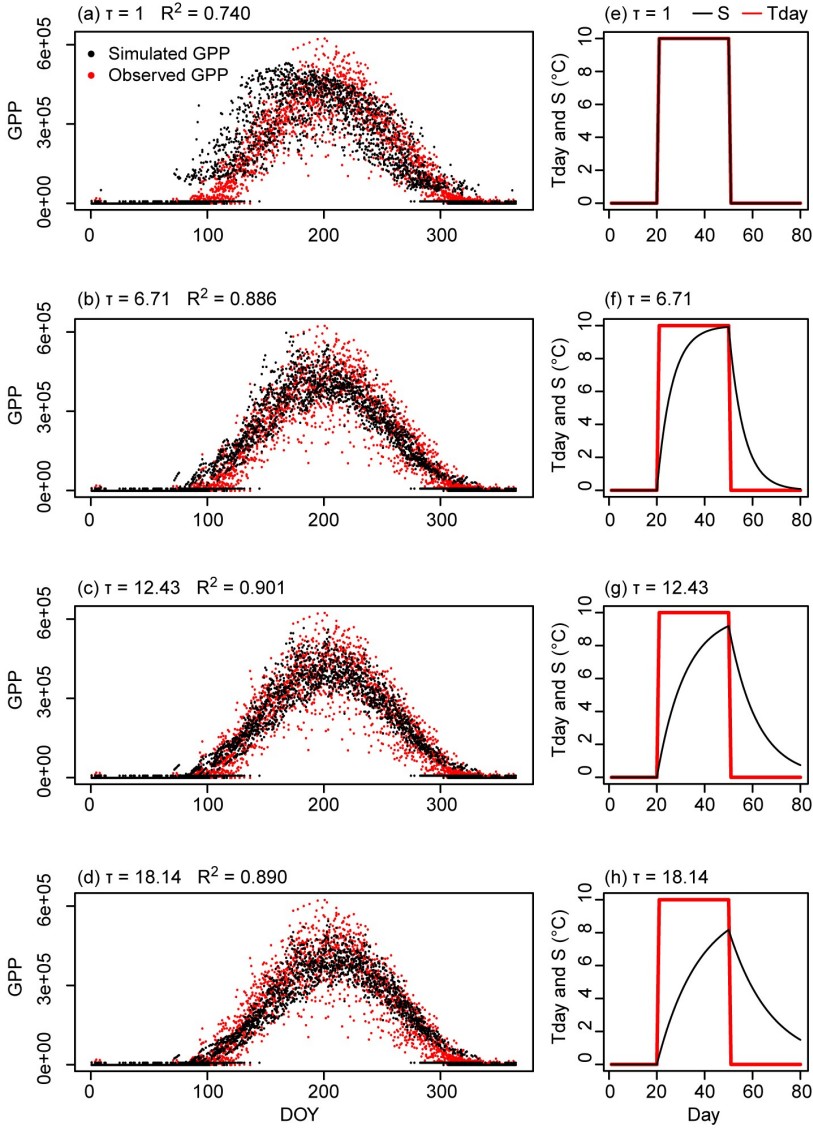

**Figure 8: Influence of the temperature transformation (*S*) on the modeled annual cycle of GPP daily values (umol C m$^{-2}$ day$^{-1}$) at the Quebec Eastern Old Black Spruce site. Only the *τ* parameter determining the *S* values was allowed to vary, while the other parameters were fixed to the values of Plausible Block GPP. (a) *τ* is 1 (*S* same as *Tday*). (b) *τ* is 6.71 (a middle value between 1 and 12.43). (c) *τ* is 12.43 (same *τ* than in Plausible Block GPP). (d) *τ* is 18.14 (a higher value than in Plausible Block GPP). The R$^2$ between observations and simulations is reported in each plot. Plots e-h show the impact of the respective *τ* values on *S* if the daily *Tday* time series corresponds to a single step of 10°C lasting 30 days.**



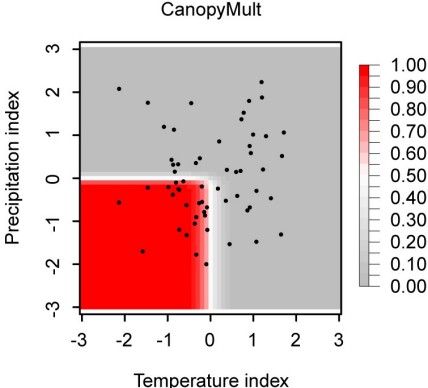

**Figure 9: Temperature and precipitation dependence of *CanopyMult* (Eq. (4); *CanopyT* and *CanopyP* are those of Plausible Block Stem), which determines the yearly canopy potential amount of carbon. Previous year mean July-August temperature indexes are on the x-axis and previous year April precipitation indexes are on the y-axis. Black dots are observed values in the central point of the region with ring width data.**

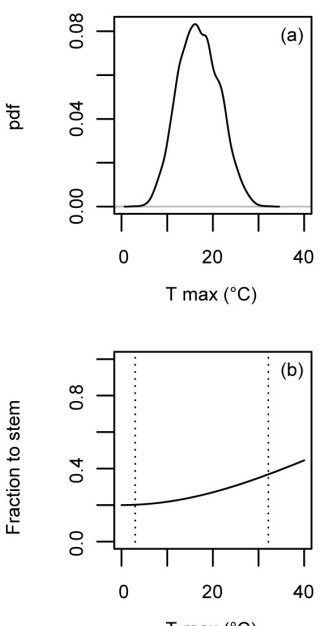

**Figure 10: Temperature dependence of the daily partition of carbon in phase 4 (growth and accumulation phase in summer) when MAIDEN is run with the parameters of Plausible Block Stem at the center of the region with ring width data in the northern Quebec taiga. (a) Probability density of daily maximum temperature values in summer. (b) Relationship between maximum temperature values and portion of carbon allocated to the stem (Eq. (6)). The vertical dashed lines show the range of maximum temperature values.**



**Table 1: Calibrated parameters.**

| Overall process | Specific Process | Eq. | Parameter | Meaning | Units | Posterior range (value in Plausible Block) |
|---|---|---|---|---|---|---|
| GPP | Temperature dependence of $Vcmax$ | 1 | $Vmax$ | Asymptote / maximum value | umol C m$^{-2}$ of leaves s$^{-1}$ | 39 / 67 (45) |
| | Temperature dependence of $Vcmax$ | 1 | $Vb$ | Slope | NA | -0.21 / -0.17 (-0.20) |
| | Temperature dependence of $Vcmax$ | 1 | $Vip$ | Inflection point | °C | 17.5 / 22.3 (18.8) |
| | Water stress level influencing the stomatal conductance | 2 | $soilb$ | Slope | NA | -0.023 / -0.008 (-0.012) |
| | Water stress level influencing the stomatal conductance | 2 | $soilip$ | Inflection point | mm | 102 / 193 (129) |
| | Acclimation to temperature of photosynthesis | 3 | $\tau$ | Needed days | days | 11.6 / 13.7 (12.4) |
| C allocation to stem | Definition of canopy maximum amount of C | 4 | $CanopyT$ | Slope of the temperature dependence | NA | 0.54 / 19.24 (6.87) |
| | Definition of canopy maximum amount of C | 4 | $CanopyP$ | Slope of the precipitation dependence | NA | 1.70 / 19.85 (16.68) |
| | Start of growing season (budburst) | NA | $GDD1$ | GDD sum threshold | °C | 56.75 / 87.05 (70.22) |
| | Start of growing season (budburst) | NA | $vegphase23$ | Day before the later start | day of the year | 161.5 / 171.0 (167.0) |
| | Start of growing season (budburst) | NA | $day23\_flex$ | Acclimation to changing GDD sums | years | 1.53 / 3.29 (2.24) |
| | Daily available C in phase 3 | NA | $Cbud$ | Storage C used by the plant | g C·m$^{-2}$ of stand·day$^{-1}$ | 1.59 / 1.86 (1.69) |
| | Partition of C in phase 3 | 5 | $h3$ | Portion allocated to canopy and roots | fraction (0-1) | 0.983 / 1.000 (0.991) |
| | Partition of C in phase 4 (stem versus storage) | 6 | $st4temp$ | Inflection point of the temperature dependence | °C | 27.53 / 59.11 (46.78) |
| | Transition from phase 4 to 5 | NA | $photoper$ | Photoperiod threshold | hours | 12.96 / 13.72 (13.41) |
| | C losses from the canopy | 7 | $PercentFall$ | Yearly canopy turnover rate | fraction (0-1) | 0.093 / 0.149 (0.143) |
| | C losses from the canopy | 7 | $OutMax$ | Approximate day of the year with maximum losses | day of the year | 154.2 / 195.0 (171.7) |
| | C losses from the canopy | 7 | $OutLength$ | Index proportional to the length of the period with losses | NA | 4.80 / 10.80 (9.91) |

**Table 2: ANOVA table for the best response function (here a combination of 4 out of the 10 tested predictors minimized the BIC) with daily GPP at EOBS as dependent variable (excluding days between November 15[th] and April 1[st]). All F values are highly significant ($p<0.001$). For precipitation, lag n indicates the sum of the daily precipitation of the week ending in day n.**

| Predictor | Pairwise correlation with GPP | ANOVA table | | | | BIC |
|---|---|---|---|---|---|---|
| | | Regression coefficient | d.f. | Variance explained | F value | |
| Maximum temperature – lag -2 days | 0.79 | 0.149 | 1 | 0.630 | 3716.16 | |
| Maximum temperature – lag -19 days | 0.69 | 0.057 | 1 | 0.049 | 291.29 | |
| Precipitation – lag -2 | 0.21 | 0.019 | 1 | 0.005 | 31.99 | |
| Precipitation – lag -0 | 0.16 | -0.013 | 1 | 0.005 | 29.90 | |
| Total variance explained | | | | 0.690 | | 5288.3 |
| Residuals | NA | NA | 1827 | 0.310 | NA | |
| Function with all 10 tested predictors | | | | 0.694 | | 5315.4 |



**Table 3: ANOVA table for the best response function (here a combination of 3 out of the 10 tested predictors minimized the BIC) with the observed mean detrended ring width series (RWhighF) as dependent variable.**

| Predictor (monthly data around the indicated day of the year) | Pairwise correlation with RWhighF | ANOVA table | | | | BIC |
|---|---|---|---|---|---|---|
| | | Regression coefficient | d.f. | Variance explained | F value | |
| Mean temperature – Previous July 28[th] | -0.60 | -0.477 | 1 | 0.355 | 39.88[***] | |
| Mean temperature – July 22[th] | 0.41 | 0.247 | 1 | 0.055 | 6.14[*] | |
| Maximum temperature – May 30[th] | 0.33 | 0.146 | 1 | 0.093 | 10.48[**] | |
| Total variance explained | | | | 0.502 | | 148.7 |
| Residuals | NA | NA | 56 | 0.498 | NA | |
| Function with all 10 tested predictors | | | | 0.568 | | 169.0 |

***($p<0.001$); **($p<0.01$); *($p<0.05$)