# Peer review of "Ecophysiological modeling of photosynthesis and carbon allocation to the tree stem in the boreal forest"

_Biogeosciences, 2017_

## Short Comment (SC1) · 29 Mar 2017

*A note upfront from the submitting person: This review was prepared by Bastian Buman, a master student in earth system science at the University of Zurich. The review was part of an exercise during a second semester master level seminar on "the biogeochemistry of plant-soil systems in a changing world", which I organize. We would like to highlight that the depth of scientific knowledge and technical understanding of these reviewers represents that of master students. We enjoyed discussing the manuscript in the seminar, and hope that our comments will be helpful for the authors.*

By refining the MAIDEN ecophysiological model (Gea-Izquierdo et al., 2015) and using it for a boreal black spruce (Picea mariana (Mill.) B.S.P.) ecosystem, the authors

(Gennaretti et al., 2017) try to shed light on factors influencing photosynthesis-carbon allocation mechanisms. They give a detailed overview of their work and the functioning of their version of the model (→ why in the boreal zone, → what improvements/changes to MAIDEN, → how were parametrizations retrieved?) Further they test model outputs (GPP and carbon allocation) with data from eddy covariance measurements and tree ring width measurements.

This work contributes both to ecophysiological models for the boreal zone as well as to the understanding of processes that govern tree stem carbon allocation. Both of which helps to understand boreal forests (in the light of climate change, carbon stock and flux estimations). The model is able to explain 90% of the daily GPP variability and 73% of the annual ring width variability (from observations). Overall the authors demonstrate that their model is working well and that it is reasonable to use this model for the boreal zone.

Title: Maybe it would make sense to remove "the climate imprint" and "North America" from the title: Ecophysiological modeling of photosynthesis and carbon allocation to the tree stem in the boreal forest. With this the title still informs about the content of the article: modeling of photosynthesis and carbon allocation and the link to tree stem growth, and as hinted in the article, the model can also be applied to other boreal forests outside of North America → attract more readers with the article?

Material and Methods: Overall well explained but tricky to get it straight. There are many factors and parts of the model explained but it would be helpful to have some kind of flowchart that explains in which order the model runs (see e.g. fig. 1 in Gea-Izquierdo et al., 2015 or Misson, 2004).

Figure 1: This figure is not optimal, although in its core it explains the MAIDEN model, text and visualization do not support each other and partly the text is not even clearly readable:

Table 1: This table displays a significant amount of the authors work but has no real

description.

One could argue that some parts in this chapter could be moved into the supplements: For extended reasoning to why something was done in whichever way: e.g. page 4, line 23 to 31 or page 5 lines 15 and 16, or page 7 sections 2.2.1 and 2.2.2.

Figure 2 is not really adding something to the paper. Why no move into supplement?

Chapter 2.2.3 Climate Data: Even though a considerable amount of work was put into acquiring climate data one might consider putting some part of this chapter into the supplements. This refers to the sentence ranging from line 9 to 13. It is an exhaustive sentence and could profit from a more detailed explanation within the supplements. This chapter is too long (especially compared to the discussion which is only half the size), having read this part, a reader must make a break or will lose attention during the next sections.

Results and Discussion: This part is – although to a lesser extent – still massive. It is quite difficult to find key aspects and concepts within the text. It would be nice to have a table (similar to Table 1), or bullet points or another form of highlighting of the key findings.

Figure 6: Is the indication "-1" really necessary when the title already states "previous year"?

Conclusion: Well written but also a bit too much text, one could remove lines 26 to 29 (page 12), (an interested researcher can always contact the authors for advice/guidance).

To me this article felt more like a very in-depth description or report of the whole thinking and working process of the authors rather than a concise research article. This is also applicable to the title. While this may sound hard, it essentially just indicates that the authors did a very good job of reasoning during their work and just need to shorten and distill everything down into the necessary. Or in other words, the current state of

the article presents a good opportunity for the authors to go back and figure out what they really would like the reader to take home from reading it.

References

Gea-Izquierdo, G., Guibal, F., Joffre, R., Ourcival, J. M., Simioni, G., and Guiot, J.: Modelling the climatic drivers determining photosynthesis and carbon allocation in evergreen Mediterranean forests using multiproxy long time series, Biogeosciences, 12, 3695–3712, doi:10.5194/bg-12-3695-2015, 2015.

Gennaretti, F., Gea-Izquierdo, G., Boucher, E., Berninger, F., Arseneault, D., and Guiot, J.: Ecophysiological modeling of the climate imprint on photosynthesis and carbon allocation to the tree stem in the North American boreal forest, Biogeosciences Discuss., 1–26, doi:10.5194/bg-2017-51, 2017.

Misson, L.: MAIDEN: A model for analyzing ecosystem processes in dendroecology, Can. J. For. Res., 34, 874–887, doi:10.1139/x03-252, 2004.

---

## Referee Comment (RC1) · Anonymous Referee #1 · 19 May 2017

General comments:

Gennaretti et al describe modifications to the ecophysiological MAIDEN model for simulation of tree ring width and gross primary productivity, including parameter estimation, and compare results with observations for the purpose of validating the model. The work builds on the extensive development of MAIDEN over man years. The presentation is admirably compact and concise. I suggest revisions to expand and improve the diagnostic study of the simulations with respect to ecophysiological controls on the predicted variables, further discussion of the extent to which the results are explained by mechanistic processes operating within the model, multiple sensitivity of the results to parameters and variables; and improvement of the presentation to separate clearly

Results, Discussion and Conclusions. The discussion of the ecophysiological and environmental controls should be central to the abstract. The publication of the comparator dataset in the supplement should be accompanied by publication of MAIDEN code in the public domain such that others may experiment with this well-studied and highly valuable model.

Specific comments:

0. Abstract:

0.1. clarify/revise results statement, what is meant by 'full spectrum', give specifics.

0.2. use of the word robust means you have done validation or out-of-sample tests of the model. Have the authors done so?

Introduction:

1. pg 2, l. 1: define 'secondary growth' at first use.

2. pg 2, l. 5: also roots; e.g. Moorcroft (2006) and description in section 2.1.

3. pg 2, l. 25: I think you mean to say that these models should be able to simulate the following observed phenomena: (i,ii,iii).

4. pg 2, l. 31-32: briefly explain for those unfamiliar with its development, why MAIDEN is an ideal model with which to work for the purposes of this study. For instance, in the sentence prior, you've noted that it was developed for Mediterranean and temperate climates; why should it be suitable for simulations in boreal climates? Here you can borrow from section 2.1 the salient descriptive points, saving for section 2.1 the description of the modifications and the experiments performed for this study. But make the argument why the model should be suitable for the present study (noting also your point that it has never been applied in 'environments mostly sensitive to cold temperatures'.

Materials and Methods:

5. pg 3, l. 21: I think you mean here: model has not been used to simulated forest growth in boreal conditions. See also note 4.

6. pg 3, l. 19-21: "Drought and water stresses are well take into account": support this statement with citations and references, but otherwise I suggest to save such statements for the Results section.

7. pg. 3, l. 29: describe how the parameter estimates are cross-validated. To determine 6 or 12 parameters simultaneously, conditioned on two variables, must require a lot of data but also out-of-sample testing [to revisit after reading Supplement 1].

[to delete? 8. pg. 4, l. 1-30: please revise to better distinguish prior formulation of MAIDEN and the modifications introduced here. What is existing, what is new here? ]

9. pg 4, l. 30: Euler's method might not be suitable in the case of a large time step, large change in the rate of change, or both; consider a Runge-Kutta solver, relatively straightforward to implement.

10. pg 5, l. 10-15, 16-20, 21-25: the determination of phenological phases seems highly specified for a modeling striving to be more ecophysiologically based (Introduction). Instead of basing these phases on correlation studies (empirical), could they be estimate from other properties of the environment, or prognostic variables within the existing model? In addition, please justify all choices of hard parameters, for instance, pg 5, l. 19, pg 6, l. 22.

Results/Discussion:

11. General remark, section 2.3: if the simulation produces a good out-of-sample or independent fit to observed predictors, then it would be good to diagnose the model: what factors are most important controls on the fidelity of the simulations? Because this is an ecophysiological modeling study, this would be much more instructive than the statistical regression analysis, although the latter may be used to support the intepretation with respect to modeled variables. Therefore, please add ecophysiological

diagnostics to this section or a new section 2.4.

12. Section 3.1, pg 9, l. 3: explain here and/or in the Table 1 caption the definition and how to interpret the series of numbers that are in the last column of the Table. What exactly do you mean by "sharp" here and on pg 11 (I think I know, but give a more objective description of what you mean for the reader).

13. pg 9, l. 7-12: "However, the ensembles of daily and annual time series retained by the MCMC sampling were not always centered on the observed time series (Fig. 5)..." Revise and expand to reflect that the simulated annual GPP values overestimate the actual GPP at low observed GPP. This will better reflect the excellent information content of this figure.

14. pg. 9, l. 13-17: Put uncertainty estimates on Fig 6 and use them in the description of results and in discussion later.

15. pg 9, l. 13-17: "The model explained about 20-30% of the 15 observed yearly RWhighF variability corresponding to correlations of 0.58-0.66 (Fig. 4b). This is a good result because simulated detrended annual GPP values (i.e. photosynthetic assimilation before any carbon allocation) had only negative R2 with RWhighF (Fig. 4c; meaning performance worse than a straight line centered on RWhighF). This suggests that the modified MAIDEN daily partition of carbon in the plant compartments significantly improved the concordance with treering observations." Although I am not sure I understand this result (and its discussion; please clarify, in mechanistic terms, why we see the results in fig 4c?): If the correlations in Fig 4c are statistically significant (estimate p-values), then this is an even more important result than described, because not only are the model improvements an important advance, but they correct a result that would otherwise produce the opposite correlation.

16. pg 9, l. 20-24: Fig 7b, here and elsewhere: it is an interesting result! But where r is given, also give effective degrees of freedom and p-value; interpret based on the p-value as statistically significant or nonsignificant.

17. General remarks on sections 2 and 3: Reorganize the content in these sections into separate Results (section 3) and Discussion Sections (new Section 4), with sub-sections as appropriate. Results are what was objectively found and will be discussed; Discussion is for interpretation of the results. As it is, Results and Discussion are entwined, but it would clarify for the reader to separate and distinguish them. I would suggest to focus the Results on the following items of interest: (1) sensitivity of the simulations to specified parameters; (2) mechanistic and regression-based diagnostics. I would then put in the Discussion the following argument: (1) The results are sensitive to parameter estimation in the following ways: .... but: (2) Comparison with independent observations suggest MAIDEN as revised produces more accurate simulation of GPP, TRW, intra-growing season dynamics ... which are (3) consistent with response function analysis, and (4?) here are some predictions made by the model/simulations, that could be tested with additional observations.

Once these revisions help to reorganize the essential content of the paper, it will be easier to evaluate the expanded ecophysiological interpretation, which I think should be more central to the main thrust of the paper than the response function analysis (Abstract; section 3.3).

18. Section 3.2: Revise the title for English; perhaps: Mechanistic diagnostics? And consolidate mechanistic results here, with their discussion in the Discussion section. Moving the Supplemental Figures that are most relevant for the central elements of the argument into the main text, and by expanding this part of the results, this may address my previous comment #11 on Section 2.3. This will help the reader understand what is going on in the simulation that explains the consistency with observations.

19. p. 9, l. 30: "..is sensitive to all..." Discuss how the sensitivity of the results to parameter estimates related to "Vcmax or thetaG, except soilb", complicate the diagnostic interpretation. Are there multiple controls that all could produce a similar result and good fit to observations?

Conclusions:

20. General reamrks on Section 4: some of the points are speculative (e.g. p 12, l. 24-26; paragraph starting at l. 30); these belong in Discussion rather than Conclusions (see note 17, last sentence).

21. I would appreciate more discussion of fig S2. Interpret the bimodal structure of the posterior probabilities; connect to the plausible value descriptions in the last column of Table 1.

22. Is MAIDEN publicly available (it was first introduced in 2004), and if not, could it be made so, to encourage experimentation in other environments, species, applications? This would be a great contribution and it would be consistent with the open data access policies of CP.

23. Fig S12: this is not isoMAIDEN as in the caption, correct?

24. Trivia: for future revisions if necessary, make line numbers cumulative rather than by page

25. Trivia: p. 9, l. 30: "..is sensitive to all..."

26. Trivia: Slight revisions for English grammar and usage: pg 3 l. 3-5; please go through entire manuscript to revise for grammar as well.

---

## Referee Comment (RC2) · Anonymous Referee #2 · 25 May 2017

General comments: Gennaretti et al. adapted MAIDEN ecophysiological forest model for North American boreal forest, which is a further application and development of MAIDEN model. It is great to see the progress of MAIDEN, and its extended application for different species and locations in tree ring and forest ecosystem research. Such process-based models will benefit our understanding in dynamic of forest carbon cycle, and its response to climate change. This paper was well written. The detailed supplementary file is also very helpful for readers to follow the key points of the article. Considering this is about further adaption of such a mature and multi-regional applied model, some more specific scientific questions/motivations might be helpful to strengthen the innovation of this research.

[Figure]

Specific comments:

1. Page3 last paragraph: About parameter tuning, there are 6 (reference: daily GPP) + 12 (reference: annual ring width) parameters tuned in this application. It is very smart of suing Bayesian optimization for such a large number of parameters tuning. However, there are only 2 references (GPP and detrended tree ring). This might have impact on the final choice of the parameter value. Some solid check about the relationship among different parameters (correlation or interaction), and sensitivity analysis for Dstem parameters is needed.

2. Page 7 line 18: Typo "Ring with" should be "Ring width"

3. Page 8 line 5: The input data for MAIDEN include daily temperature and precipitation, as well as CO2. Is the solar-related parameter needed, e.g. sunshine hour, cloud cover fraction? If not, please briefly demonstrate how photosynthesis was estimated.

4. Page 8 last paragraph: In the third step of this research (evaluation of the model performance), the indirect comparison between variance explanation (R2) of model simulation and climate response function was applied. It would be helpful to verify the model performance by showing the same climate response function analysis for the model simulation, e.g. combined Table 2 and 3 for the both observed and simulated GPP and Dstem. And it would be even more convincing to show the moving correlation analysis (figure 6) between simulated Dstem and monthly climate.

5. Page 9 line 3: R2 was widely used in this model-data comparison. a) The calculation method for R2 is needed here. b) Was model (parameter) was tuned using the same whole observation, or only a portion of the observation? A bit curious about the not small negative value of R2.

6. Page 9 line 15: It makes sense that the annual GPP has a very poor correlation with ring width. One of the obvious reason is the definition of "annual" and the carbon carry over from previous year, which is the stored carbon in MAIDEN. I guess "annual"

in the paper means the calendar year (Jan to Dec). It would be very useful to check the correlation between GPP in effective carbon year (or growth year, e.g. previous July to current June or from previous Phase 4 to Phase 3) and ring width observation.

7. Page 12 line 23: Does increasing $CO_2$ contributed to this positive relation between summer temperature and Dstem? Is there any $CO_2$ fertilisation signal in both the simulated Dstem and the observed ring width?

8. Page 11 line 1: Does this stored carbon include previous year's stored carbon? What would happen, if the stored carbon was used up, e.g. carbon was stored very little during previous year?

9. Page2 line 5: "compartments" mean "component"?

10. Figure 4: The method of calculating $R^2$ need to be specified, either in the method part or the figure caption part. Is there any constraint when $R^2$ was calculated, especially for the negative $R^2$?

11. Figure 5: Please enlarge the scatter plot for both the Daily GPP and the Annual GPP

12. Figure 6: Is it possible to add the same moving correlation for simulated Dstem?

13. Figure 7: Please enlarge the scatter plot for both the detrended and raw Dstem.

14. Figure 9: The information about the colour scale is needed in the caption.

15. Table 1: It would be good to add the prior range in this table

―――――――――――――――――――――

---

## Author Comment (AC1) · 29 Jun 2017

We are very happy that all reviewers appreciated our work and that our paper has already been used for teaching purposes (see the interactive comment #1). We also acknowledge the reviewers for their constructive comments that improved the paper. You will find attached a point-by-point response for all comments of the reviewers, as well as the new version of the manuscript and of the supplements.

Please also note the supplement to this comment:
https://www.biogeosciences-discuss.net/bg-2017-51/bg-2017-51-AC1-supplement.zip

---

## Author Comment (AC2) · 29 Jun 2017

We have answered to the reviewer in our comment "AC1"

---

## Author Comment (AC3) · 29 Jun 2017

We have answered to the reviewer in our comment "AC1"

---

## Author Response (AR1)

We are very happy that all reviewers appreciated our work and that our paper has already been used for teaching purposes (see the interactive comment #1). We also acknowledge the reviewers for their constructive comments that improved the paper. Here below you will find a point-by-point response for all comments of the reviewers.

Reviewer #1:

Comment 1
0. Abstract:
0.1. clarify/revise results statement, what is meant by 'full spectrum', give specifics.

Response
We reformulated the sentence:
P 1 L 17 "MAIDEN explains 90% of the observed daily gross primary production variability, 73% of the annual ring width variability and 20-30% of its high frequency component (i.e. when decadal trends are removed)."

Comment 2
Introduction:
1. pg 2, l. 1: define 'secondary growth' at first use.

Response
We added the definition.
P 1 L 30 "secondary growth is the increase of the girth of the plant roots and stems"

Comment 3
2. pg 2, l. 5: also roots; e.g. Moorcroft (2006) and description in section 2.1.

Response
We modified the sentence and added the reference:
P 2 L 3 "Indeed, carbon allocated in different tree components (e.g. canopy, stem or roots) has a specific function and is stored for a different length of time (Moorcroft, 2006)"

Comment 4
3. pg 2, l. 25: I think you mean to say that these models should be able to simulate the following observed phenomena: (i,ii,iii).

Response
We modified as suggested.
P 2 L 23 "Such models should be able to simulate the following observed phenomena:…"

Comment 5
4. pg 2, l. 31-32: briefly explain for those unfamiliar with its development, why MAIDEN is an ideal model with which to work for the purposes of this study. For instance, in the sentence prior, you've noted that it was developed for Mediterranean and temperate climates; why should it be suitable for simulations in boreal climates? Here you can borrow from section 2.1 the salient descriptive points, saving for section 2.1 the description of the modifications and the experiments performed for this study. But make the argument why the model should be suitable for the present study (noting also your point that it has never been applied in 'environments mostly sensitive to cold temperatures'.

Response
We added a sentence:
P 2 L 29 "MAIDEN offers an ideal framework to analyze the impact of introducing in the model relevant processes for carbon assimilation and allocation in temperature sensitive boreal trees. Indeed, the model simultaneously simulates the course of photosynthesis and sets different

phenological phases to determine the allocation of carbon to different plant compartments in a dynamical manner."

Comment 6
Materials and Methods:
5. pg 3, l. 21: I think you mean here: model has not been used to simulated forest growth in boreal conditions. See also note 4.

Response
We modified as suggested.
P 3 L 16 "Up to now, the model has never been used to simulate forest growth in boreal conditions."

Comment 7
6. pg 3, l. 19-21: "Drought and water stresses are well take into account": support this statement with citations and references, but otherwise I suggest to save such statements for the Results section.

Response
We deleted the sentence.

Comment 8
0. Abstract:
0.2. use of the word robust means you have done validation or out-of-sample tests of the model. Have the authors done so?
7. pg. 3, l. 29: describe how the parameter estimates are cross-validated. To determine 6 or 12 parameters simultaneously, conditioned on two variables, must require a lot of data but also out-of-sample testing [to revisit after reading Supplement 1].

Response
We added a cross-validation exercise of parameter values:
Figs. S5, S9 and S10.
Supplement P 2 L 12 "The robustness of the parameters' posterior distributions was tested with a cross-validation exercise. Firstly, we compared the parameters' posterior densities, when the optimization was executed on the full period with observed data, to those obtained with half data (Figs. S5 and S9). However, we have to recall that in total we have 2920 observed daily data between 2003 and 2010 to optimize the 6 parameters influencing the GPP, and only 61 observed ring width annual data between 1950 and 2010 to optimize the 12 parameters influencing Dstem. Subsequently, the distributions of the parameters influencing Dstem were also compared to those obtained independently with data from specific sites (the used black spruce ring width data comes from five different riparian forests; Fig. S10)."

Comment 9
[to delete? 8. pg. 4, l. 1-30: please revise to better distinguish prior formulation of MAIDEN and the modifications introduced here. What is existing, what is new here? ]

Response
We added a sentence:
P 4 L 8 "The computations of Vcmax and θg used here are identical to those of the prior formulation of MAIDEN (Gea-Izquierdo et al., 2015)."

Comment 10
9. pg 4, l. 30: Euler's method might not be suitable in the case of a large time step, large change in the rate of change, or both; consider a Runge-Kutta solver, relatively straightforward to implement.

Response
We understand the reviewer comment. However, the Euler's method is a particular case of the Runge–Kutta family of methods and we think that a different solver will not change significantly our results and interpretations concerning the need to take into account acclimation of photosynthesis to temperature for boreal trees.

Comment 11
10. pg 5, l. 10-15, 16-20, 21-25: the determination of phenological phases seems highly specified for a modeling striving to be more ecophysiologically based (Introduction). Instead of basing these phases on correlation studies (empirical), could they be estimate from other properties of the environment, or prognostic variables within the existing model? In addition, please justify all choices of hard parameters, for instance, pg 5, l. 19, pg 6, l. 22.

Response
We modified the text to better justify our choices.
P 5 L 9 "Based on previous studies on black spruce forests (Girardin et al., 2016; Ols et al., 2016; Mamet and Kershaw, 2011), we modified the model to consider the effect of the previous year April precipitation and July-August temperature likely influencing the length and the thermal-hydraulic stress of the previous growing season, respectively. Previous year climate conditions of specific months are known to influence shoot extension of boreal trees likely because they control accumulation of resources in the buds (Salminen and Jalkanen, 2005)."

P 5 L 22 "In this way, $AlloCcanopy_j$ may vary between the 70% and the 100% of $MaxCcanopy$ as in the previous version of the model (Gea-Izquierdo et al., 2015)."

P 6 L 20 "The value 0.8 was chosen to force a minimum threshold of C allocation to the stem in this phase (at least 20%) and to guarantee the correspondence between the inflection point and the temperature where roughly 50% of $CT_i$ is allocated to the stem."

Comment 12
Results/Discussion:
11. General remark, section 2.3: if the simulation produces a good out-of-sample or independent fit to observed predictors, then it would be good to diagnose the model: what factors are most important controls on the fidelity of the simulations? Because this is an ecophysiological modeling study, this would be much more instructive than the statistical regression analysis, although the latter may be used to support the interpretation with respect to modeled variables. Therefore, please add ecophysiological diagnostics to this section or a new section 2.4.
18. Section 3.2: Revise the title for English; perhaps: Mechanistic diagnostics? And consolidate mechanistic results here, with their discussion in the Discussion section. Moving the Supplemental Figures that are most relevant for the central elements of the argument into the main text, and by expanding this part of the results, this may address my previous comment #11 on Section 2.3. This will help the reader understand what is going on in the simulation that explains the consistency with observations.

Response
We modified the title of section 3.2 as suggested. We believe that mechanistic results for GPP are already shown and discussed in depth (see Figs. 6, S4, and S12 to S18). We consolidated results and discussion of mechanistic rules for Dstem. Figs. 7 and 8, illustrating how specific processes impact the MAIDEN simulations, were indeed modified to show how the parameter selection of those processes alters the correlation with observed data. We also added Figs. S6, S7, S11, S20 in addition to the already existing Figs. S8, S19 and S21.

Comment 13
12. Section 3.1, pg 9, l. 3: explain here and/or in the Table 1 caption the definition and how to interpret the series of numbers that are in the last column of the Table. What exactly do you mean

by "sharp" here and on pg 11 (I think I know, but give a more objective description of what you mean for the reader).

Response
We added a sharpness definition in the text:
P 8 L 20 "The posterior distributions of the parameters were quite sharp (Fig. S4; Table 1; by sharpness we mean the shrinking of the distribution relative to the prior acceptable range toward a posterior distribution with a well-defined, narrow peak). Sharp distributions with small posterior ranges relative to the prior ones indicate sensitive parameters."

We added how to interpret prior and posterior ranges in the caption of Table 1:
"Small posterior ranges relative to the prior ones indicate sensitive parameters."

Comment 14
13. pg 9, l. 7-12: "However, the ensembles of daily and annual time series retained by the MCMC sampling were not always centered on the observed time series (Fig. 5)..." Revise and expand to reflect that the simulated annual GPP values overestimate the actual GPP at low observed GPP. This will better reflect the excellent information content of this figure.

Response
We modified as suggested.
P 8 L 29 "However, the ensembles of daily and annual time series retained by the MCMC sampling were not always centered on the observed time series (Fig. 3), in particular the simulated annual GPP values often underestimated the actual GPP especially at low observed GPP."

Comment 15
14. pg. 9, l. 13-17: Put uncertainty estimates on Fig 6 and use them in the description of results and in discussion later.

Response
We modified the figure adding the thresholds of significance ($p<0.05$).

Comment 16
15. pg 9, l. 13-17: "The model explained about 20-30% of the 15 observed yearly RWhighF variability corresponding to correlations of 0.58-0.66 (Fig. 4b). This is a good result because simulated detrended annual GPP values (i.e. photosynthetic assimilation before any carbon allocation) had only negative R2 with RWhighF (Fig. 4c; meaning performance worse than a straight line centered on RWhighF). This suggests that the modified MAIDEN daily partition of carbon in the plant compartments significantly improved the concordance with treering observations." Although I am not sure I understand this result (and its discussion; please clarify, in mechanistic terms, why we see the results in fig 4c?): If the correlations in Fig 4c are statistically significant (estimate p-values), then this is an even more important result than described, because not only are the model improvements an important advance, but they correct a result that would otherwise produce the opposite correlation.

Response
Correlations between GPP and RWhighF are positive (r=0.3, see text over Fig. 2c), such as those between Dstem and RWhighF (r=0.65, see text over Fig. 2b). However, $R^2$ values between GPP and RWhighF are negative (see Fig. 2c), while those between Dstem and RWhighF are positive (see Fig. 2b). To clarify this point, we have shown the equation to compute the $R^2$ (Eq. 8) and a comparison between GPP, Dstem and RWhighF (Figs. S6 and S7).

Comment 17
16. pg 9, l. 20-24: Fig 7b, here and elsewhere: it is an interesting result! But where r is given, also give effective degrees of freedom and p-value; interpret based on the p-value as statistically significant or nonsignificant.

Response
We now provide all df and p-values of correlation coefficients in the text.

Comment 18
17. General remarks on sections 2 and 3: Reorganize the content in these sections into separate Results (section 3) and Discussion Sections (new Section 4), with subsections as appropriate. Results are what was objectively found and will be discussed; Discussion is for interpretation of the results. As it is, Results and Discussion are entwined, but it would clarify for the reader to separate and distinguish them. I would suggest to focus the Results on the following items of interest: (1) sensitivity of the simulations to specified parameters; (2) mechanistic and regression-based diagnostics. I would then put in the Discussion the following argument: (1) The results are sensitive to parameter estimation in the following ways: .... but: (2) Comparison with independent observations suggest MAIDEN as revised produces more accurate simulation of GPP, TRW, intra-growing season dynamics ... which are (3) consistent with response function analysis, and (4?) here are some predictions made by the model/simulations, that could be tested with additional observations.
Once these revisions help to reorganize the essential content of the paper, it will be easier to evaluate the expanded ecophysiological interpretation, which I think should be more central to the main thrust of the paper than the response function analysis (Abstract; section 3.3).

Response
We did not modified the paper as the reviewer suggested because the proposed modifications are much more than a reorganization and because the current structure is already well defined according to us:

3. Results and Discussion
3.1 GPP and tree-ring growth variability explained by MAIDEN
3.2 Mechanistic diagnostics
3.3 Comparison between MAIDEN and response functions
3.4 Limits and error sources of the study

Furthermore, the ecophysiological interpretation is already central in the Results and Discussion section (subsection 3.3 on the comparison with response functions is only approximately 1/6 of section 3).

Comment 19
19. p. 9, l. 30: "..is sensitive to all..." Discuss how the sensitivity of the results to parameter estimates related to "Vcmax or thetaG, except soilb", complicate the diagnostic interpretation. Are there multiple controls that all could produce a similar result and good fit to observations?
21. I would appreciate more discussion of fig S2. Interpret the bimodal structure of the posterior probabilities; connect to the plausible value descriptions in the last column of Table 1.

Response
The impact of the parameters on the simulations is already shown on Figs. 6 and S14-S18. We added and modified some text related to this reviewer's comment:
P 8 L 20 "The posterior distributions of the parameters were quite sharp (Fig. S4; Table 1; by sharpness we mean the shrinking of the distribution relative to the prior acceptable range toward a posterior distribution with a well-defined, narrow peak). Sharp distributions with small posterior ranges relative to the prior ones indicate sensitive parameters. This means that the model posterior probability (i.e. model plausibility) increased significantly with the specific values of the selected parameters retained by the MCMC sampling. The slightly bimodal structures of the

posterior distributions of Vmax, Vb and Vip were likely a consequence of their significant cross-correlations (Table S1). However, the posterior distributions of these three parameters were robust and consistent even when the Bayesian optimization was executed on independent periods (Fig. S5)."

Comment 20
Conclusions:
20. General reamrks on Section 4: some of the points are speculative (e.g. p 12, l. 24-26; paragraph starting at l. 30); these belong in Discussion rather than Conclusions (see note 17, last sentence).

Response
We moved the indicated paragraph in the discussion.

Comment 21
22. Is MAIDEN publicly available (it was first introduced in 2004), and if not, could it be made so, to encourage experimentation in other environments, species, applications? This would be a great contribution and it would be consistent with the open data access policies of CP.
General comment
publication of MAIDEN code in the public domain such that others may experiment with this well-studied and highly valuable model

Response
The used MAIDEN version will be made publicly available upon the paper acceptance on "Figshare". The DOI will be updated at the next step of the reviewing process.
P 13 L 26 "The used MAIDEN version is publicly available on "Figshare": DOI: to be obtained."

Comment 22
23. Fig S12: this is not isoMAIDEN as in the caption, correct?

Response
We corrected our mistake.

Comment 23
24. Trivia: for future revisions if necessary, make line numbers cumulative rather than by page

Response
We used the Copernicus "Word" template.

Comment 24
25. Trivia: p. 9, l. 30: "..is sensitive to all..."

Response
We corrected the mistake.

Comment 25
26. Trivia: Slight revisions for English grammar and usage: pg 3 l. 3-5; please go through entire manuscript to revise for grammar as well.

Response
We revised the sentence:
P 3 L 2 "This comparison allows to verify that the process-based ecophysiological model satisfactorily reproduces the variability of the observed data and that its simulations keep robust relationships with the most significant climate variables."

###############################################################

**Comment 1**
1. Page3 last paragraph: About parameter tuning, there are 6 (reference: daily GPP) + 12 (reference: annual ring width) parameters tuned in this application. It is very smart of suing Bayesian optimization for such a large number of parameters tuning. However, there are only 2 references (GPP and detrended tree ring). This might have impact on the final choice of the parameter value. Some solid check about the relationship among different parameters (correlation or interaction), and sensitivity analysis for Dstem parameters is needed.

Response
We added a cross-correlation analysis of the parameters' values (Tables S1 and S2; see also discussion of these tables in the main text), a sensitivity analysis of some central Dstem parameters (Figs. 7, 8 and S20), and a cross-validation of parameters' distributions (Figs. S5, S9 and S10; see also discussion in the main text).

**Comment 2**
2. Page 7 line 18: Typo "Ring with" should be "Ring width"

Response
We corrected the mistake.

**Comment 3**
3. Page 8 line 5: The input data for MAIDEN include daily temperature and precipitation, as well as CO2. Is the solar-related parameter needed, e.g. sunshine hour, cloud cover fraction? If not, please briefly demonstrate how photosynthesis was estimated.

Response
MAIDEN can use two different meteorological input data: (1) a complete dataset composed of daily temperature, precipitation, CO2, radiation, relative humidity and wind speed; (2) a reduced dataset composed of daily temperature, precipitation and CO2. We used MAIDEN with the reduced input data and letting the model estimate the other variables as explained by Misson (2004).

In the case of radiation, Misson (2004) explains:
"Climatic driving variables are daily minimum and maximum temperatures, precipitation, global radiation, and atmospheric vapor pressure deficit (Table 1). Since radiation and humidity variables are usually not available for large temporal and spatial scale applications, we coupled the MT-CLIM model (Running et al. 1987) to MAIDEN to estimate these variables from observations of daily maximum and minimum temperatures and precipitation. In MT-CLIM, humidity estimates are based on the fact that daily minimum temperature is usually very close to the dew point (Running et al. 1987). Radiation estimates are based on the fact that the diurnal temperature range is closely related to the daily mean atmospheric transmittance (Running et al. 1987; Thornton et al. 2000)."

We modified our text to better clarify the input data required by MAIDEN while we refer to Misson (2004) for a more in detail description of the estimation of the micrometeorological covariates:
P 3 L 8 "Starting from daily minimum-maximum air temperature, precipitation and CO2 atmospheric concentration (these are the minimum required input variables which are completed by radiation, relative humidity and wind speed when additional meteorological data are available; Misson, 2004), MAIDEN models the phenological and meteorological controls on GPP and carbon allocation (Fig. 1; see also flowcharts in Misson, 2004 and Gea-Izquierdo et al., 2015)."

The MAIDEN code will also be freely available upon the paper acceptance and the readers can directly verify on the code the used equations. The DOI will be updated at the next step of the reviewing process.

P 13 L 25 "The used MAIDEN version is publicly available on "Figshare": DOI: to be obtained."

Running, S. W., Nemani, R. R., and Hungerford, R. D.: Extrapolation of synoptic meteorological data in mountainous terrain and its use for simulating forest evapotranspiration and photosynthesis, Can. J. Forest Res., 17, 472-483, doi:10.1139/x87-081, 1987.

Thornton, P. E., Hasenauer, H., and White, M. A.: Simultaneous estimation of daily solar radiation and humidity from observed temperature and precipitation: an application over complex terrain in Austria, Agr. Forest Meteorol., 104, 255-271, doi:https://doi.org/10.1016/S0168-1923(00)00170-2, 2000.

Comment 4
4. Page 8 last paragraph: In the third step of this research (evaluation of the model performance), the indirect comparison between variance explanation (R2) of model simulation and climate response function was applied. It would be helpful to verify the model performance by showing the same climate response function analysis for the model simulation, e.g. combined Table 2 and 3 for the both observed and simulated GPP and Dstem. And it would be even more convincing to show the moving correlation analysis (figure 6) between simulated Dstem and monthly climate.
12. Figure 6: Is it possible to add the same moving correlation for simulated Dstem?

Response
We modified the figure (Fig. 4) as suggested, showing the moving correlations for simulated Dstem.

Comment 5
5. Page 9 line 3: R2 was widely used in this model-data comparison. a) The calculation method for R2 is needed here. b) Was model (parameter) was tuned using the same whole observation, or only a portion of the observation? A bit curious about the not small negative value of R2.
10. Figure 4: The method of calculating R2 need to be specified, either in the method part or the figure caption part. Is there any constraint when R2 was calculated, especially for the negative R2?

Response
A cross-validation of parameters' distributions can be found on Figs. S5, S9 and S10 (see also discussion in the main text).
We added the $R^2$ computation method.
P 7 L 5 "The proportion of the observed variability explained by MAIDEN was evaluated with the coefficient of determination (R2), which compares the performance of simulated time series relative to that of straight horizontal lines centered on the data:

$$R^2 = 1 - \frac{\sum_i (Obs_i - Sim_i)^2}{\sum_i (Obs_i - \overline{Obs})^2}$$

Comment 6
6. Page 9 line 15: It makes sense that the annual GPP has a very poor correlation with ring width. One of the obvious reason is the definition of "annual" and the carbon carry over from previous year, which is the stored carbon in MAIDEN. I guess "annual" in the paper means the calendar year (Jan to Dec). It would be very useful to check the correlation between GPP in effective carbon year (or growth year, e.g. previous July to current June or from previous Phase 4 to Phase 3) and ring width observation.

Response
We checked the reviewer suggestion and produced a new figure (Fig. S6).

Comment 7
7. Page 12 line 23: Does increasing CO2 contributed to this positive relation between summer temperature and Dstem? Is there any CO2 fertilisation signal in both the simulated Dstem and the observed ring width?

Response
We added Figure S11 and some text.
P 9 L 17 "Indeed, the positive trend in response to the warming of the last few decades was well captured by the model simulations of stem increments, which included some CO2 fertilization contribution (Fig. S11)."

Comment 8
8. Page 11 line 1: Does this stored carbon include previous year's stored carbon? What would happen, if the stored carbon was used up, e.g. carbon was stored very little during previous year?

Response
Yes, the stored carbon include previous year's stored carbon. If no stored carbon is used in the budburst phase the correlations between Dstem and RWhighF drop down (Fig. S20). We modified some text accordingly:
P 10 L 33 "In phase 3, corresponding to Budburst, a portion of the available carbon simulated by MAIDEN comes from stored non-structural carbohydrates from the current and previous years (parameter Cbud; see Table 1). In our case, Cbud was quantified as about 1.69 g C•m-2 day-1 (Fig. S8f) and this remobilization improves the correlations between Dstem and RWhighF (Fig. S20). However, the Cbud selection was also sensitive to the period and the site used in the optimization (Figs. S9 and S10)."

Comment 9
9. Page2 line 5: "compartments" mean "component"?

Response
We replaced "compartments" by "component".

Comment 10
11. Figure 5: Please enlarge the scatter plot for both the Daily GPP and the Annual GPP

Response
We did the modification.

Comment 11
13. Figure 7: Please enlarge the scatter plot for both the detrended and raw Dstem.

Response
We did the modification.

Comment 12
14. Figure 9: The information about the colour scale is needed in the caption.

Response
We added in the caption (Fig. 7) the requested information: "unitless multiplier"

Comment 13
15. Table 1: It would be good to add the prior range in this table

Response
We added the prior range

################################################################

Interactive comment #1:

Comment 1
Title: Maybe it would make sense to remove "the climate imprint" and "North America" from the title: Ecophysiological modeling of photosynthesis and carbon allocation to the tree stem in the boreal forest. With this the title still informs about the content of the article: modeling of photosynthesis and carbon allocation and the link to tree stem growth, and as hinted in the article, the model can also be applied to other boreal forests outside of North America ! attract more readers with the article?

Response
We agree with this suggestion and modified the title.

Comment 2
Material and Methods: Overall well explained but tricky to get it straight. There are many factors and parts of the model explained but it would be helpful to have some kind of flowchart that explains in which order the model runs (see e.g. fig. 1 in Gea- Izquierdo et al., 2015 or Misson, 2004).
Figure 1: This figure is not optimal, although in its core it explains the MAIDEN model, text and visualization do not support each other and partly the text is not even clearly readable:

Response
We referred to the already published flowcharts and increased readability of Figure 1 reducing box transparency.
P 3 L 11 "(Fig. 1; see also flowcharts in Misson, 2004 and Gea-Izquierdo et al., 2015)"

Comment 3
Table 1: This table displays a significant amount of the authors work but has no real description.

Response
We added a more complete description.

Comment 4
One could argue that some parts in this chapter could be moved into the supplements: For extended reasoning to why something was done in whichever way: e.g. page 4, line 23 to 31 or page 5 lines 15 and 16, or page 7 sections 2.2.1 and 2.2.2.
This chapter is too long (especially compared to the discussion which is only half the size), having read this part, a reader must make a break or will lose attention during the next sections.

Response
We reduced several sections of our Materials and Methods and moved the information in the Supplement S2.

Comment 5
Figure 2 is not really adding something to the paper. Why no move into supplement?

Response
We moved Figs. 2 and 3 in the supplements (Figs. S2 and S3)

Comment 6
Chapter 2.2.3 Climate Data: Even though a considerable amount of work was put into acquiring climate data one might consider putting some part of this chapter into the supplements. This refers to the sentence ranging from line 9 to 13. It is an exhaustive sentence and could profit from a more detailed explanation within the supplements.

Response
We reduced the chapter and moved some information in the Supplement S2.

Comment 7
Results and Discussion: This part is – although to a lesser extent – still massive. It is quite difficult to find key aspects and concepts within the text. It would be nice to have a table (similar to Table 1), or bullet points or another form of highlighting of the key findings.

Response
Key findings are already highlighted by the figures:
Figs. 2 to 5 = Performance of MAIDEN in reproducing observed data
Figs. 6 to 8 = Impact of key model adaptations on the simulations and on the correspondence with observed data.

Comment 8
Figure 6: Is the indication "-1" really necessary when the title already states "previous year"?

Response
We deleted "-1" (Fig. 4).

Comment 9
Conclusion: Well written but also a bit too much text, one could remove lines 26 to 29 (page 12), (an interested researcher can always contact the authors for advice/guidance).

Response
We removed these lines.

[revised manuscript text omitted]

---

## Author Response (AR2)

We are happy that the reviewer found our first revision to be almost satisfactory. Here below you will find a point-by-point response for each of the reviewer's residual comments.

###

Reviewer #1:

1. Comment 18: I still think it highly desirable to separate Results (what was found) from Discussion (how the authors interpret the results in support of their argument). At the very least (at the Editor's discretion), please check the structure of each subsection for this clear separation. For example, the first paragraph could clearly be Results and the second paragraph is clearly Discussion of those results. That will clarify these elements for the reader.

Response
We reorganized the paper separating Results and Discussion.

2. For a model validation study such as this, it is also valuable to make testable predictions or forecasts. It would be good to collect these predictions (for instance, of the applicability of the model to predictions of forest growth and allocation dynamics in other boreal forests; for the prediction of the response of the studied boreal forest growth and allocation dynamics under past or future environmental change) into one subsection that is clearly Discussion. This might include the Bayesian estimation of parameters, and here Fig S5, S9 would be useful results to discuss. For example, some parameters appear bimodal with all data but become unimodal with subsets; others show the reverse phenomenon. Why do the authors think this is, and do the uncertainties in the posterior parameter pdfs themselves affect their interpretations? (e.g. Fig S8)

Response
We introduced a new section with some model predictions:
"4.3 Some model predictions
It is possible to use the new optimized MAIDEN version to predict forest growth and allocation dynamics of the studied boreal forests under future environmental change. In the study area, the daily maximum temperature, the daily minimum temperature and the precipitation should increase at the 2050 horizon of about 2.3°C, 4.3°C and 12%, respectively (Guay et al., 2015). If we modify the used climatic data (Section 2.2.3) by these median changes and we fix the CO2 concentration at the 2010 level, the median increase of the annual GPP and Dstem values simulated by MAIDEN for the studied forests is of 43 and 68%, respectively (Fig. S22). We have to recall that the ring width data used for the optimization of MAIDEN come from lake riparian trees and that these results are too optimistic for more water limited boreal sites."

We also grouped all discussion about parameter interpretation on a new 4.1 section. In this section, we decided to focus on: (1) the shrinking of the parameters' posterior distributions relative to the prior acceptable ranges; (2) the bimodal structures of some parameters; (3) some clear inconsistencies when optimizing on different periods or sites; (4) the impact of parameters' uncertainties on our interpretations; (5) the in depth discussion of some important parameters.
"4.1 Parameter interpretation
The posterior distributions of the parameters were quite sharp (Figs. S4 and S5; Table 1; by sharpness we mean the shrinking of the distribution relative to the prior acceptable range toward a posterior distribution with a well-defined, narrow peak). Sharp distributions with small posterior ranges relative to the prior ones indicate sensitive parameters. This means that the model posterior probability (i.e. model plausibility) increased significantly with the specific values of the selected parameters retained by the MCMC sampling. The slightly bimodal structures of the posterior distributions of *Vmax*, *Vb* and *Vip* were likely a consequence of their significant cross-correlations (Table S1). However, the posterior distributions of these three parameters were robust and consistent even when the Bayesian optimization was executed on independent periods (Fig. S19). The optimization of some parameters controlling Dstem (the three related to the start of the growing season and *Cbud*) was sensitive to the choice of the period and the site in

the cross-validation exercise (Figs. S20 and S21) likely as a result of the short length of the available observed data (61 yearly RWhighF values) and of some significant cross-correlation coefficients (Table S2). However, in all cases, the uncertainties in the parameters' posterior distributions (Figs. S4 and S5) did not affect our interpretations because the MAIDEN simulations were extremely consistent whatever the selected block of parameters (see Figs. 3 and 5).

The interpretation of some parameters needs special care. For example, those controlling the negative impact of both previous year April precipitation and July-August temperature values on canopy development, which may be explained by the following reasons: warm previous Aprils with infrequent late snowfalls may accelerate snowmelt and the start of the previous growing season, allowing optimal reserve accumulation during the previous year which would influence tree performance the following growing year. This mechanism may be significant especially if we do not observe high temperatures limiting soil water availability and reserve accumulation during the previous summer (Girardin et al., 2016). It has already been shown that shoot elongation of boreal conifers is determined by climate conditions during bud formation (Salminen and Jalkanen, 2005). However, for Scots pine, previous summer temperatures are positively correlated with shoot elongation, while in our case, the opposite process was simulated and the simulations were even more sensitive to the values of the CanopyT temperature dependent parameter than to those of the CanopyP precipitation dependent parameter (Fig. 7b-e). Clearly, we need more data on canopy development and shoot elongation to verify the model results."

3. The discussion of testable predictions would highlight another important element that belongs in the new section 3.4. This is the extent to which the model structure might be overly tuned to the simulation of the target with the multiple fixed parameters described in 2.1.2. This should be discussed in general terms: identify the potential problem and then allow for the possibility of their estimation rather than specification. Of course this would require more tuning and more input data, but this is the price we pay for model complexity.

Response
We added a sentence in section 4.4:
"Second, some fixed parameters are present in the MAIDEN code (see Eqs. 4 and 6). These parameters might be potentially tuned but their specification is justified in Section 2.1.2 and limits additional parameter tuning."

4. Please review the revised/new text, including table and figure captions, for grammar; e.g. "Warmer temperature corresponded to a greater portion of carbon allocated to the stem and less to non-structural carbohydrates (Fig. 8a-b), being the simulations highly sensitive to the st4temp parameter (Fig. 8c-d)." --> "Warmer temperatures corresponded to a greater amount of carbon allocated to the stem and less to non-structural carbohydrates (Fig. 8a-b), because the simulations are highly sensitive to the ..."

Response
We revised the sentence and more carefully checked our text for grammar mistakes.

###

In the next pages you will find a marked-up manuscript version where new insertions are in red and moved sections are in green.

[revised manuscript text omitted]

---

## Author Response (AR3)

Please, find here below a point-by-point response to the editor's comments.

###

1. First of all, I think it is good to separate the results and the discussion section as recommended by the referee. You have done that, but the beginning of your discussion still sounds like a results section. Could you please add some introductory sentences about general findings and give the reader some guidance for the discussion section.

Response
We have added some introductory sentences
"In this study, the MAIDEN model was successfully modified to consider important processes for boreal tree species and to improve the simulation of the coupling between photosynthesis and carbon allocation to the stem in boreal forests. Because we used a Bayesian optimization procedure, we start the following discussion with the interpretation of the parameters' posterior distributions (section 4.1) and of the simulation uncertainties (section 4.2). Subsequently, some model predictions at the 2050 horizon are presented to identify the likely response of the studied boreal forests under future environmental change (section 4.3). Finally, we conclude by illustrating factors that may potentially influence the obtained results (section 4.4)."

2. Second, the manuscript generally needs language revisions by a native speaker.

Response
We have asked to an English editing service (American Journal Experts) to polish the language of our manuscript.

###

In the next pages a marked-up manuscript version can be found where new insertions are in red.

[revised manuscript text omitted]